# Myosin 7a is required for maintaining the transducing stereocilia and for force transmission to the MET channel during cochlear hair cell development

Anna Underhill[1], Samuel Webb[1], Fiorella C. Grandi[2] , Adam J. Carlton[1] , Jing-Yi Jeng[1], Stuart L. Johnson[1,3], Corné J. Kros[4]  and Walter Marcotti[1,3] 

[1]*School of Biosciences, University of Sheffield, Sheffield, UK*

[2]*Sorbonne Université, INSERM, Institute de Myologie, Centre de Recherche en Myologie, Paris, France*

[3]*Neuroscience Institute, University of Sheffield, Sheffield, UK*

[4]*School of Life Sciences, University of Sussex, Falmer, Brighton, UK*

Handling Editors: Kim Barrett & Pawel Ferdek

The peer review history is available in the Supporting Information section of this article (https://doi.org/10.1113/JP289623#support-information-section).

**Abstract figure legend** Developmental changes in the hair-bundle structure of outer hair cells (OHCs) and inner hair cells (IHCs) were examined in control and *shaker-1* mutant ($Myo7a^{Sh1/Sh1}$) mice, which carry a spontaneous missense mutation in the *Myo7a* gene that produces a non-functional MYO7A protein. In the mutants, hair cells progressively lose the two shortest rows of stereocilia, leading to a corresponding loss of mechanoelectrical transduction (MET) in adulthood. Before MET is lost, the mutation causes a reduction in the resting open probability ($P_o$, purple arrows) of the MET channel, although more mature cells eventually regain a normal resting $P_o$. Notably, in immature mutant hair cells, depletion of cholesterol [with methyl-$\beta$-cyclodextrin (M$\beta$CD)] or PIP$_2$ [with phenylarsine oxide (PAO)] from the lipid

A. Underhill and S. Webb contributed equally to the work.

The Journal of Physiology

bilayer restored resting $P_o$ (curved blue arrows), indicating that MYO7A is unlikely to be required directly for setting the resting $P_o$ of the MET channel.

**Abstract** *Shaker-1* mice carry a spontaneous missense mutation in *Myo7a* (*Myo7a$^{Sh1}$*) that interferes with the motor function of MYO7A. Mutation in the orthologous gene in humans causes syndromic (Usher 1B) or non-syndromic forms of deafness. In hair cells, MYO7A is expressed throughout the stereocilia, where it has been implicated in tip-link tensioning required for gating the mechanoelectrical transducer (MET) channel and setting its resting open probability ($P_o$). The *Myo7a$^{Sh1}$* mutation progressively dysregulated the height of shorter stereocilia rows from the end of the first postnatal week onwards, associated with reduced MET current amplitude and hearing loss. Noise exposure exacerbated stereocilia dysfunction in *Myo7a$^{Sh1/Sh1}$* mice. Following the onset of maturation, hair cells from *Myo7a$^{Sh1/Sh1}$* mice showed normal resting $P_o$ and calcium sensitivity of the MET channel. In immature *Myo7a$^{Sh1/Sh1}$* hair cells, the resting $P_o$ was very small or absent in comparison to control cells, but it was restored by changing the membrane lipid bilayer fluidity or thickness by depleting cholesterol or PIP$_2$. Bundle stiffness in immature IHCs was not affected by the absence of functional MYO7A but decreased after their onset of maturation in both genotypes. Expression of a subset of genes was affected similarly in immature *Myo7a$^{Sh1/Sh1}$* mice and in adult *Myo7a* conditional knockout mice, indicating a common response pathway in *Myo7a*-deficient mice. This study reveals that MET channel gating might differ depending on hair cell developmental stage, and MYO7A is likely to influence, albeit indirectly, force transmission via the lipid bilayer to the MET channel and maintenance of the shorter rows of transducing stereocilia.

(Received 27 June 2025; accepted after revision 22 January 2026; first published online 12 February 2026)
**Corresponding author** W. Marcotti: School of Biosciences, University of Sheffield, Sheffield S10 2TN, UK. Email: w.marcotti@sheffield.ac.uk

## Key points

- *Shaker-1* mice carry a spontaneous missense mutation in the *shaker-1* gene (*Myo7a$^{Sh1}$*) that interferes with the motor function of MYO7A, a protein expressed in hair-cell stereocilia.
- The absence of functional MYO7A (*Myo7a$^{Sh1/Sh1}$* mice) caused a progressive dysregulation in the height of the shortest two rows of stereocilia and the consequent loss of mechanoelectrical transduction (MET) current.
- Although immature hair cells from *Myo7a$^{Sh1/Sh1}$* mice exhibited a markedly reduced resting open probability of their MET channels, this was restored upon maturation or following depletion of cholesterol or PIP$_2$ from the lipid bilayer.
- Hair-bundle stiffness was affected in immature inner hair cells from *Myo7a$^{Sh1/Sh1}$* mice, suggesting that MYO7A is not required for establishing the resting tension of the tip links gating the MET channels.
- We conclude that MYO7A is crucial for the structural integrity of the MET complex and transport of key proteins required to transfer forces efficiently from the lipid bilayer to the MET channel.

## Introduction

Unconventional myosins are a class of motor proteins that couple ATP hydrolysis with mechanical force, allowing them to move along actin-based filaments (Goode et al., 2000). These myosins are expressed ubiquitously in eukaryotic cells and are involved in a wide range of cellular functions, including actin organization, transport, anchoring and tension sensing (Hartman et al., 2011). MYO7A is one of these unconventional myosins and is part of the MyTH4-FERM sub-family that is specialized in the transport of actin-regulatory proteins and other factors to the tip of actin-based protrusions (Weck et al., 2017). In the mammalian auditory organ, the cochlea, MYO7A is highly expressed in the actin-rich stereociliary hair bundles (e.g. Hasson et al., 1997; Rzadzinska et al., 2004; gEAR: https://umgear.org/), which are microvilli-like structures that project from the apical

surface of the sensory hair cells (Tilney et al., 1992). Mutations in *MYO7A* cause syndromic (Usher 1B) or non-syndromic recessive deafness in humans (Liu et al., 1997; Weil et al., 1995, 1997), highlighting the crucial role of this protein in hearing.

The stereociliary bundles atop the hair cells convert mechanically induced acoustic stimuli within the cochlear partition into electrical signals that are then relayed to the central auditory pathway (Qiu & Müller, 2022). In the mouse cochlea, each hair bundle is composed of three rows of stereocilia, each of which is anchored to the hair cells via their basal taper region (Park & Bird, 2023). Stereocilia within a hair bundle are interconnected by several extracellular linkages (Goodyear et al., 2005; Tilney et al., 1992), one of these being the tip link, which is formed by cadherin 23 (CDH23) and protocadherin 15 (PCDH15) at the upper and lower end, respectively (Kazmierczak et al., 2007; Siemens et al., 2004). Tip links are thought to transmit force generated during hair-bundle displacement to gate the mechano-electrical transducer (MET) channel via a direct interaction between PCDH15 and the mechanotransduction complex, which is located at the tips of the shorter two rows of stereocilia (Beurg et al., 2009). At the other end of the tip link, CDH23 interacts with the actin cytoskeleton of the taller stereocilia via the upper tip-link density (UTLD), which is believed to be formed by MYO7A and the adaptor proteins harmonin and sans (Caberlotto et al., 2011; Grillet et al., 2009; Grati & Kachar, 2011). Several observations have led to the hypothesis that harmonin and sans provide the scaffolding required for MYO7A to act as the force generator that, by tensioning the tip links, keeps the open probability of the MET channels within the most sensitive region of their operating range (Qiu & Müller, 2022). Although MYO7A has been shown to cluster at the UTLD (Grati & Kachar, 2011), it is also highly expressed throughout the stereocilia and cell body (Hasson et al., 1997; Li et al., 2020; Michalski et al., 2007; Senften et al., 2006; Underhill et al., 2025), where it performs several roles. For example, it is responsible for localizing various proteins at the base of the stereocilia (e.g. ADGRV1, PDZD7: Moreland & Bird, 2022) and for transporting the capping protein twinfilin-2 to the top of the shorter rows of stereocilia (Peng et al., 2009; Rzadzinska et al., 2009).

In this study, we investigated the role of MYO7A using the *shaker-1* mouse, which has a spontaneous missense mutation in the *shaker-1* gene ($Myo7a^{Sh1}$) that interferes with its motor function (Gibson et al., 1995). The same *Sh1* mutation underlies Usher syndrome type 1B in humans (Weil et al., 1995). Although the hair bundles of *shaker-1* mice seem to form normally, at least during the initial stages of development, they deteriorate progressively, leading to substantial hearing loss by the age of post-natal day (P)15 (Mikaelian & Ruben, 1964; Self et al., 1998). Using a combination of functional, morphological and molecular approaches, we propose that MYO7A is unlikely to be directly responsible for setting the resting open probability ($P_o$) of the MET channel in hair cells before and after their onset of maturation, further supporting recent findings in adult *Myo7a* conditional knockout mice (Underhill et al., 2025).

## Methods

### Ethics approval

The use of mice was licensed by the UK Home Office under the Animals Act 1986 (PCC8E5E93 and PP1481074) and was approved by the University of Sheffield Ethical Review Committee (180626_Mar). Littermate mice of either sex were used for all the experimental work. Mice had free access to food and water and a 12 h–12 h light–dark cycle. *Shaker-1* mice ($Myo7a^{sh1}$), which were obtained from MRC Mary Lyon Centre (Harwell Campus, UK), were maintained on the original background (85% CBA and 15% mixed) (Gibson et al., 1995). For cochlear hair cell experiments, both male and female mice were killed by cervical dislocation followed by decapitation. Mice used for auditory brainstem responses (ABRs) were anaesthetized

**Anna Underhill** studied for her BSc in Physiological Sciences at Newcastle University. She went on to receive her MSc in Clinical Neuroscience from the University of Edinburgh, followed by a PhD in Walter Marcotti's laboratory at the University of Sheffield. Her PhD research investigated the role of myosin 7a in mechanotransduction in the mammalian cochlea, with the aim of understanding the link between myosin 7a and deafness in Usher syndrome. **Samuel Webb** is a postdoctoral researcher within the Hearing Research Group at the University of Sheffield. He earned his PhD in auditory neuroscience from Manchester Metropolitan University in 2019 and now focuses on uncovering the biological mechanisms underlying auditory function and dysfunction. His present research examines how ageing and noise exposure drive auditory decline, aiming to identify key mechanisms that can be targeted to improve long-term hearing health.

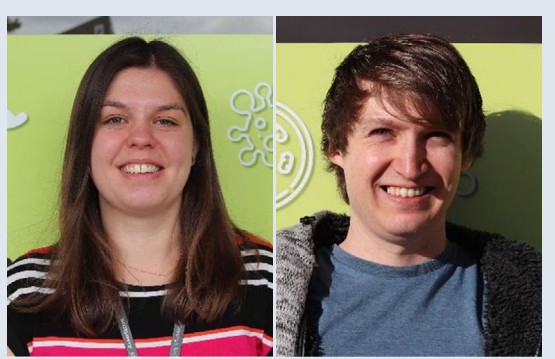

using an intraperitoneal injection of ketamine (100 mg/kg body weight, Fort Dodge Animal Health, Fort Dodge, IA, USA) and xylazine (10 mg/kg, Rompun 2%, Bayer HealthCare LLC, NY, USA). After confirming a complete loss of the retraction reflex via toe pinch, mice were placed in a soundproof chamber for ABR recordings. At the end of the *in vivo* experiments, mice were either killed by cervical dislocation or allowed to recover from anaesthesia by intraperitoneal injection of atipamezole (1 mg/kg). Mice under recovery from anaesthesia were returned to their cage, placed on a thermal mat, and monitored over the following 2–5 h. All animal experiments performed in this study comply with the journal's policies regarding animal experiments (https://physoc.onlinelibrary.wiley.com/hub/animal-experiments).

### Preparation for *ex vivo* cochlear tissue

The entire inner ear of the mouse was placed in a Petri dish containing the following extracellular solution (mM): 135 NaCl, 5.8 KCl, 1.3 $CaCl_2$, 0.9 $MgCl_2$, 0.7 $NaH_2PO_4$, 5.6 D-glucose and 10 HEPES–NaOH. Amino acids, vitamins and sodium pyruvate (2 mM) were added from concentrates (Thermo Fisher Scientific, UK). The final solution had a pH of 7.48 and osmolality of ∼308 mosmol/kg. A nylon mesh attached to a stainless-steel ring was used to immobilize the dissected sensory epithelium of the cochlea at the bottom of a microscope chamber. To maintain a healthy *ex vivo* preparation, the above extracellular solution was continuously perfused through the microscope chamber using a peristaltic pump (Cole-Palmer, UK). The microscope chamber was then mounted on the stage of an upright microscope (Olympus BX51, Japan; Leica DMLFS, Germany; Nikon FN1, Germany) equipped with Nomarski differential interference contrast (DIC) optics, 60× or 64× water immersion objectives and a 15× eyepiece. All *ex vivo* experiments were performed at room temperature (20–24°C).

### Whole-cell patch-clamp electrophysiology

Mechanoelectrical transducer (MET) currents were recorded using an Optopatch amplifier (Cairn Research Ltd, UK). Patch pipettes were pulled from soda glass capillaries, which had a typical resistance in extracellular solution of 2–3 MΩ, and contained (mM): 135 CsCl, 2.5 $MgCl_2$, 1 EGTA–CsOH, 2.5 $Na_2ATP$, 10 sodium phosphocreatine and 5 HEPES–CsOH (pH 7.3). Data acquisition was controlled by pClamp software using a Digidata 1440A (Molecular Devices, USA). Recordings were low-pass filtered at 2.5 kHz (eight-pole Bessel), sampled at 5 kHz and stored on a computer for off-line

analysis (Clampfit, Molecular Devices, USA; Origin: OriginLab, USA). MET currents were plotted at specific membrane potentials that were corrected off-line for the liquid junction potential of −4 mV, which was measured between electrode and bath solutions. MET current recordings and bundle displacement measurements were taken from the apical-coil region of the cochlea (around 9–12 kHz).

The MET currents were elicited by displacing the stereociliary bundles of both inner hair cells (IHCs) and outer hair cells (OHCs) using a mechanical stimulus (50 Hz sinusoids, filtered at 1 kHz, eight-pole Bessel) or a symmetrical step protocol alternating from positive to negative voltages, which were delivered through a pipette by a custom-made fluid-jet system (Corns et al., 2014; Underhill et al., 2025).

The fluid-jet pipette, which was pulled from borosilicate glass, contained extracellular solution (see above). Prior to the positioning of the fluid jet by the hair bundles, any steady-state pressure was removed via a side-port connected to the fluid-jet chamber, which was required to provide a reliable measurement of the resting $P_o$ of the MET channels (Corns et al., 2014, 2018). In experiments designed to assess the contribution of cholesterol to the MET current, methyl-$\beta$-cyclodextrin (M$\beta$CD; Sigma-Aldrich, C4555) was added to both the extracellular solution bathing the cochlea and the fluid-jet pipette, as previously described (Beurg et al., 2024). M$\beta$CD was dissolved in the usual extracellular solution. Given that M$\beta$CD treatment progressively reduces the MET current amplitude, we selected concentrations (1.0–3.5 mM; most recordings at 2 mM) and pre-incubation times (12–36 min) that allowed us to measure its effects on the MET channel resting $P_o$ prior to significant current reduction, with the higher concentrations needing less time to have an effect. In a few experiments, phenylarsine oxide (PAO; Sigma-Aldrich, P3075), a phosphatidylinositol-4-kinase blocker that prevents the synthesis of phosphatidylinositol-4,5-bisphosphate (PIP$_2$; Effetz et al., 2017), was used in the extracellular solution as described for M$\beta$CD. PAO was dissolved in DMSO using sonication (stock concentration, 100 mM), then diluted 1:1000 in extracellular solution (final concentration, 100 μM), which was bath applied for a period of 11–32 min prior to recordings.

### Hair-bundle displacement and stiffness measurement

For bundle stiffness measurements, the hair bundles of IHCs were displaced with voltage steps (see above) of increasing driver voltages (DV) up to ±5 V in ±0.5 V increments. The fluid jet was positioned 15 μm away from the IHC hair bundles in all recordings at both P10 and P13.

Hair bundles were imaged with a fast camera (VEO 610L, Phantom) using the Phantom Camera Control software (Phantom) as previously described (Underhill et al., 2025). In brief, hair-bundle position was acquired at 20,000 frames/s. Individual frames were saved as TIFFs and converted to a stack for processing using ImageJ. From the stacks, we produced two-dimensional kymographs, where the *x*-plane represents time (each pixel/frame = 50 μs) and the *y*-plane space (each pixel = 15.27 nm). Steady-state hair-bundle stiffness was calculated using the fluid-jet linear fluid velocity calibrated against a carbon fibre (diameter, 8 μm) (Corns et al., 2014; Underhill et al., 2025) and by modelling the hair-bundle dimensions as previously described (Géléoc et al., 1997). The estimation of the width and height of the hair bundles was performed using an upright microscope (Nikon FN1, Germany) that we also used for electrophysiology experiments, which was equipped with a 60× water immersion objective, a 2× magnification lens, 15× eyepieces and a calibrated *z*-axis (focus).

### Immunofluorescence imaging

For antibody staining, the inner ear was dissected out from the mouse head and immersed in a solution containing 4% paraformaldehyde in PBS (pH 7.4) for 20 min at room temperature. The inner ears were then washed in PBS three times (10 min each) before dissecting out neurosensory epithelia. The isolated epithelia were then incubated for 1 h at room temperature with PBS supplemented with normal goat or horse serum (5%) and Triton X-100 (0.5%). This was followed by primary antibody immunostaining (overnight at 37°C) in PBS containing 1% serum and 0.5% Triton X-100. Primary antibodies used were as follows: rabbit anti-PCDH15-CD2 (dilution 1:500; gift from Robert Fettiplace), mouse-IgG2a anti-PMCA (1:400, MA3-914; ThermoFisher); rabbit-IgG anti-ESPNL (dilution 1:100; gift from Peter Bar-Gillespie) (Ebrahim et al., 2016). After the overnight incubation, samples were washed three times with PBS and incubated with the secondary antibodies (species appropriate Alexa Fluor) for 1 h at 37°C. F-Actin was stained with Texas Red-X phalloidin (1:400; ThermoFisher, T7471) within the secondary antibody solution. Samples were mounted in Vectashield (H-1000). Fluorescence images were taken from the apical-coil region of the cochlea (∼12 kHz), which corresponds to that used for the MET current recordings, using a ZEISS LSM980 Airyscan confocal microscope (Wolfson Light Microscope Facility at the University of Sheffield). Image stacks were processed with Fiji ImageJ software. At least three mice for each genotype were used for each experiment.

### Noise exposure

Awake and unrestrained mice were exposed to noise in a wire cage suspended at the centre of a custom-made wooden box as previously described (Holme & Steel, 2004). Broadband noise (1–16 kHz) was generated using Audacity software (Audacity v.3.2.1). The stimuli were amplified to 100 dB sound pressure level (SPL) using a W audio TPX 650 amplifier connected to a U-Phoria UMC22 audio interface and played through a speaker (Beyma TPL200/S, Spain) positioned at the top of the soundproof box. The sound pressure varied by <2 dB SPL across the suspended wire cage used to house mice during noise exposure. Noise calibration to the target SPL and frequency was performed prior to each noise exposure session using a low-noise microphone probe system (ER10B+, Etymotic, USA) connected to an oscilloscope (Picoscope 2000 series, Pico Technology, UK). Mice exposed to noise (2 h) were continuously monitored using a camera for potential adverse reactions.

### Auditory brainstem responses

Auditory brainstem responses (ABRs) were recorded by placing anaesthetized mice of either sex on a heated mat (37°C) positioned inside a soundproof chamber (MAC-3 acoustic chamber, IAC Acoustic, UK). For sound delivery, the loudspeaker (MF1-S, Multi Field Speaker, Tucker-Davis Technologies, USA) was positioned 10 cm in front of the pinna of the mouse. The loudspeaker was calibrated daily with a low-noise microphone probe system (ER10B+, Etymotic, USA). ABR data were recorded using three subdermal electrodes: the active electrode was positioned halfway between the two pinnae on the vertex of the cranium; the reference and earth electrodes were placed under the skin behind the pinna of each ear. Experiments were performed using customized software (Ingham et al., 2011) driving an RZ6 auditory processor (Tucker-Davis Technologies, USA). ABR thresholds were estimated based on the lowest sound level at which any recognizable feature of the waveform was visible. Stimulus sound pressure levels were presented in 5 dB SPL increments, each representing an average of 256 repetitions. Tone bursts were 5 ms in duration, with a 1 ms on/off ramp time presented at a rate of 42.6/s.

### Scanning electron microscopy

For scanning electron microscopy, the dissected cochlea of at least three mice for each genotype was gently perfused with fixative (2.5% v/v glutaraldehyde in 0.1 M sodium cacodylate buffer plus 2 mM CaCl$_2$, pH 7.4) for 1–2 min through the round window using a small pipette tip. To avoid any build-up of pressure within the cochlea, a small hole was made in the cochlear bone to allow the fixative to

flow out during the perfusion. After this initial perfusion, the cochlea was incubated in the above fixative for 2 h at room temperature on a rotating shaker. After the fixation, the sensory epithelium was exposed by removing the bone around the cochlear apex, then immersed in 1% osmium tetroxide in 0.1 M cacodylate buffer for 1 h. The following step required osmium impregnation, which was performed by incubating the cochlea twice in solutions of saturated aqueous thiocarbohydrazide (20 min) alternating with 1% osmium tetroxide in buffer (2 h) (OTOTO technique; Furness & Hackney, 1986). The samples were then dehydrated through an ethanol series and critical point dried using $CO_2$ as the transitional fluid (Leica EM CPD300, Germany). Finally, the treated cochleae were mounted on specimen stubs using conductive silver paint (Agar Scientific, Stansted, UK). The apical coil of the sensory epithelium (9–12 kHz region) was examined using an FEI Inspect F scanning electron microscope (Sorby Centre for Electron Microscopy, University of Sheffield). Measurements of the stereocilia height in both OHCs and IHCs were performed as previously described (Dunbar et al., 2019). In brief, hair-bundle images were taken at 10 kV (working distance, 10 mm) with a ±5° tilt between them (−5°, 0 and +5°). For each measure (e.g. length from one stereocilium of the tallest row), images are taken on a first micrograph and measured again on the corresponding ±5°-tilted repeat micrograph.

### RNA isolation and library preparation for RNA-sequencing

The sensory epithelia of both cochleae from each control and *shaker-1* mouse were dissected out in DNase-free ice-cold PBS 1× and immediately snap frozen in liquid nitrogen. When ready for RNA extraction, tissue was thawed on ice, and the tissue from four or five mice was combined in one tube. RNA was extracted using RNeasy Plus Micro Kit (Qiagen) according to the manufacturer's instructions, and RNA was eluted into 15 µl of $dH_2O$. RNA quantity was established using a Nanodrop spectrophotometer, and the RNA integrity number (RIN) was calculated using a BioAnalyzer, which was >9.1 in all samples. Preparation of the mRNA library was performed using poly A enrichment and sequenced on the Illumina NovaSeq sequencer using paired-end 150 bp reads.

### RNA-sequencing analysis

The sequencing libraries were processed using the nf-core RNA pipeline (Ewels et al., 2020; https://nf-co.re/rnaseq/usage), and reads were mapped to the mouse genome (mm10). Salmon (Patro et al., 2017) was used to determine the resulting gene counts. Enriched Gene Ontology (GO) terms and pathways in the list of differentially expressed genes were identified using DESeq2 (Love et al, 2014). Metascape (Zhou et al., 2019), Reactome (Gillespie et al., 2022) and StringDB (https://string-db.org/) were used to query for enriched GO terms and pathways in the list of differentially expressed genes. Isoform analysis was carried out with the programs Stringtie (Frazee et al., 2015) and ballgown (Frazee et al., 2015) for visualization.

### Statistical analysis

Statistical comparisons were made by Student's *t* test or the Mann–Whitney *U* test (when a normal distribution could not be assumed). For multiple comparisons, one-way ANOVA followed by a suitable *post hoc* test was used for normally distributed data; otherwise, the Kruskal–Wallis test with Dunn's *post hoc* test was used. For ABR pure tone experiments, owing to the presence of 'not found' values (i.e. above the upper threshold limit of our equipment), the non-parametric aligned-ranks transformation two-way ANOVA was used. A value of $P < 0.05$ was selected as the criterion for statistical significance. The assumptions of parametric testing were verified prior to analysis. Normality of data distribution was assessed using the Shapiro–Wilk test. Average values are quoted in text and figures as means ± SD. Animals of either sex were randomly assigned to the different experimental groups. Animals used were taken from several cages and breeding pairs over a period of several months. No statistical methods were used to define sample size, which was defined based on similar previously published work from our laboratory.

### Results

### The missense mutation in the *Myosin 7a* gene leads to highly disorganized hair-bundle morphology and hearing loss in adult mice

The functional consequences of the missense mutation in *shaker-1* ($Myo7a^{Sh1}$) mice were investigated using ABRs, which reflect the electrical activity of the afferent neurons innervating the IHCs. At P18, which is ~1 week after the onset of hearing in most altricial rodents (~P12; Mikaelian & Ruben, 1965; Shnerson & Pujol, 1982), ABR thresholds for click sound stimuli were already highly elevated in $Myo7a^{Sh1/Sh1}$ compared with control $Myo7a^{+/Sh1}$ mice (Fig. 1*A*). By 1 month of age (P31–P38), $Myo7a^{Sh1/Sh1}$ mice had undetectable click ABR thresholds within the dB SPL range tested (Fig. 1*A*). When hearing function was tested using frequency-specific pure tone burst stimuli, ABR thresholds were also significantly elevated in $Myo7a^{Sh1/Sh1}$ compared with

control ($Myo7a^{+/Sh1}$) mice at both P18 and P31–P38 (Fig. 1*B* and *C*).

At 1 month of age, the number of stereocilia in both IHCs and OHCs of $Myo7a^{Sh1/Sh1}$ mice was significantly reduced compared with those in control mice ($P < 0.0001$, two-way ANOVA for both hair cell types; Fig. 1*D–F*).

The hair bundles of $Myo7a^{Sh1/Sh1}$ mice were completely devoid of stereocilia in row 3 (Fig. 1*E* and *F*). Although ~50% of stereocilia in row 2 were present in both hair cell types (Fig. 1*F*), they appeared abnormally short or long compared with those in aged-matched $Myo7a^{+/Sh1}$ mice (Fig. 1*D–F*). These morphological abnormalities

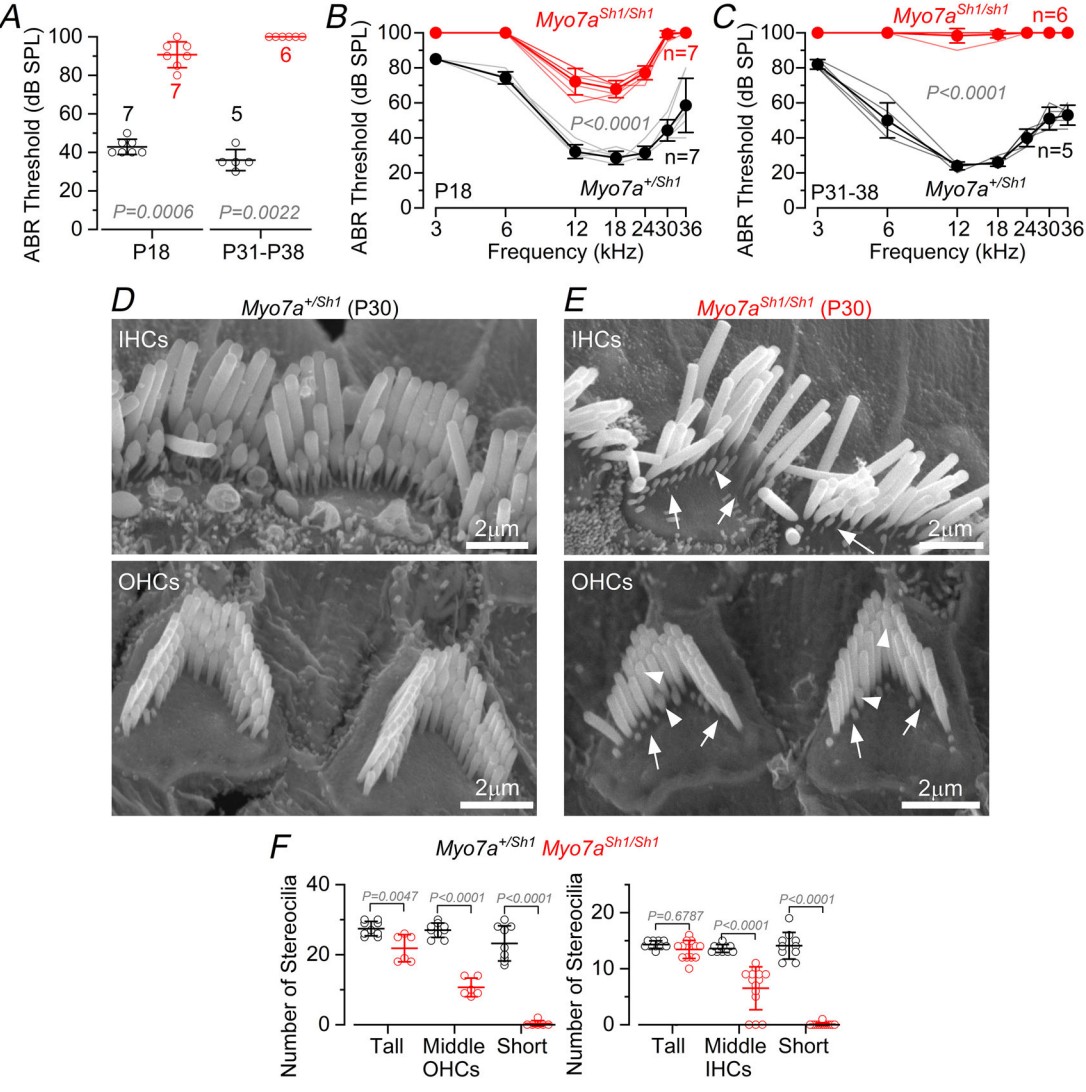

**Figure 1. Auditory brainstem response thresholds and hair bundle structure in $Myo7a^{Sh1}$ mice**
*A*, average ABR thresholds for click stimuli recorded from control $Myo7a^{+/Sh1}$ (black) and $Myo7a^{Sh1/Sh1}$ (red) mice at postnatal day 18 (P18) and P31–P38. At both ages, click thresholds were significantly elevated in $Myo7a^{Sh1/Sh1}$ compared with control mice. Statistical comparisons shown in panels are from the Mann–Whitney *U* test. *B* and *C*, ABR thresholds for frequency-specific pure tone burst stimuli at 3, 6, 12, 18, 24, 30 and 36 kHz recorded from control and littermate $Myo7a^{Sh1/Sh1}$ mice at P18 (*B*) and P31–P38 (*C*). The number of mice tested for each genotype is shown next to the symbols. Statistical values shown in *B* and *C* were obtained from non-parametric aligned ranks transformation two-way ANOVA. *D* and *E*, scanning electron microscope images showing IHC (top) and OHC (bottom) hair bundles from the apical coil of the cochlea of P30 control (*D*) and $Myo7a^{Sh1/Sh1}$ (*E*) mice. In *E* ($Myo7a^{Sh1/Sh1}$ mice), arrows point to the missing stereocilia in the middle and shortest rows, and arrowheads indicate abnormally short or long stereocilia. *F*, number of stereocilia in the tall, middle and short rows in the hair bundles of both outer hair cells (OHCs, left) and inner hair cells (IHCs, right) of $Myo7a^{+/Sh1}$ and $Myo7a^{Sh1/Sh1}$ mice. The number of hair cells/mice investigated is as follows: OHCs, 9/5 $Myo7a^{+/Sh1}$ and 6/4 $Myo7a^{Sh1/Sh1}$; IHCs, 10/7 $Myo7a^{+/Sh1}$ and 14/8re $Myo7a^{Sh1/Sh1}$. Statistical tests were performed using two-way ANOVA with Šídák's multiple comparisons *post hoc* test. Values in *A–C* and *F* are shown as the mean ± SD. Abbreviations: ABR, auditory brainstem response; IHC, inner hair cell; OHC, outer hair cell; P, postnatal day.

contribute to the highly elevated ABR thresholds in $Myo7a^{Sh1/Sh1}$ mice, because rows 2 and 3 are the transducing stereocilia containing the MET channels at their tip (Beurg et al., 2009).

### Resting MET current in immature OHCs from $Myo7a^{Sh1/Sh1}$ mice

MET currents were recorded by displacing the OHC hair bundles of P5 mice using a 50 Hz sinusoidal force stimulus from a piezo-driven fluid jet (see Methods). Saturating hair-bundle displacement towards the taller row of stereocilia (i.e. in the excitatory direction) elicited a large inward MET current at negative membrane potentials in both control ($Myo7a^{+/Sh1}$) and littermate $Myo7a^{Sh1/Sh1}$ P5 mice (Fig. 2*A*–*C*). By stepping the membrane potential from $-124$ mV to more depolarized values in 20 mV increments, the MET current decreased in size at first, then reversed near 0 mV in both genotypes (Fig. 2*C*). At the most hyperpolarized ($-124$ mV) and depolarized ($+96$ mV) voltage steps tested, the maximum size of the MET current was comparable in OHCs from both genotypes (Fig. 2*D*). Despite the similar MET current size, the resting $P_o$ of the MET channel (e.g. resting MET current in the absence of hair-bundle deflection), which is responsible for the current flowing in the absence of mechanical stimulation (Fig. 2*E*) (Corns et al., 2014), was very small in P5 OHCs from $Myo7a^{Sh1/Sh1}$ mice at both $-124$ and $+96$ mV (Fig. 2*F*).

Calcium entry via the resting MET current is known to induce a degree of adaptation, which leads to the partial closure of the MET channels and a reduction of resting MET current (Corns et al., 2014). We tested whether $Ca^{2+}$-dependent adaptation of the MET current was affected in *shaker-1* mice by performing some recordings with an intracellular solution containing the fast $Ca^{2+}$ chelator BAPTA (Fig. 3), instead of EGTA (Fig. 2). We found that by increasing the intracellular BAPTA concentration from 0.1 to 5 mM, which strongly buffers $[Ca^{2+}]_i$, the resting MET current in control OHCs was significantly augmented, but this manipulation had no or minimal effect on $Myo7a^{Sh1/Sh1}$ mice (Fig. 3*A*–*C*).

Overall, these results suggest that MYO7A appears to contribute to the resting MET current in OHCs prior to the onset of functional maturation, which occurs towards the end of the first postnatal week in mice (about P7–P8; Abe et al., 2007; Marcotti & Kros, 1999).

### Resting $P_o$ and MET current size in OHCs from $Myo7a^{Sh1}$ mice following their onset of maturation

Towards the end of the first postnatal week, some of the OHCs from $Myo7a^{Sh1/Sh1}$ mice begin to show shorter stereocilia in the second and third row compared with

those in control mice (Fig. 4*A*). This initial morphological defect is likely to explain the slightly reduced MET current recorded in OHCs from P7–P8 $Myo7a^{Sh1/Sh1}$ mice compared with controls (Fig. 4*B* and *C*). Despite the smaller MET currents, some of the P7–P8 OHCs from $Myo7a^{Sh1/Sh1}$ mice began to exhibit a normal resting MET current (Fig. 4*B* and *D*). This result also agrees with recent findings from mice that lack the canonical long isoform of MYO7A ($Myo7a$-$\Delta C$ mouse), in which OHCs showed a normal resting MET current at P7–P8 (Li et al., 2020).

During the second postnatal week, OHC hair-bundle defects associated with the *shaker-1* missense mutation

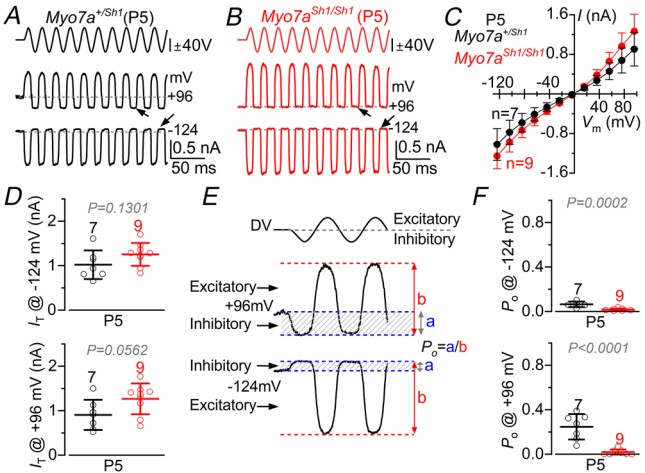

**Figure 2. Mechanoelectrical transduction in immature outer hair cells from $Myo7a^{Sh1}$ mice**

*A* and *B*, examples of saturating MET currents recorded from apical-coil OHCs of P5) control (*A*) and littermate $Myo7a^{Sh1/Sh1}$ (*B*) mice. Sinusoidal force stimuli applied to the hair bundles (50 Hz, see Methods) elicited large MET currents at membrane potentials between $-124$ and $+96$ mV in 20 mV increments (holding potential, $-84$ mV). For clarity, only the steps to the most hyperpolarized ($-124$ mV) and depolarized ($+96$ mV) voltage steps are shown. The sinusoids above the traces indicate the DV stimulus to the fluid jet, with positive deflections of the DV being excitatory (i.e. towards the taller stereocilia). The arrows indicate the size of the MET current available at rest (see *E* below). *C*, average peak-to-peak MET current as a function of the OHC membrane potential. $P = 0.6434$, two-way ANOVA. *D*, maximal size of the MET current recorded from the OHCs of both genotypes at $-124$ mV (top) and $+96$ mV (bottom). *E*, schematic representation of the method used to calculate the resting MET channel $P_o$ at both $-124$ mV (bottom) and $+96$ mV (top) membrane potentials. The value of $P_o$ was calculated by dividing the MET current available at rest, which is the current during inhibitory bundle deflection minus the holding current (a, grey dashed region), by the total MET current (b, red double-headed arrows). The sine wave above the traces is the DV stimulus to the fluid jet. *F*, resting $P_o$ of the MET channel in OHCs from both genotypes at $-124$ mV (left) and $+96$ mV (right). Statistical values shown in *D* and *F* were obtained using Student's *t* test. Data in *C*, *D* and *F* are shown as the mean ± SD. Number of OHCs/mice in *C*, *D* and *F*: $Myo7a^{+/Sh1}$, 9/6 and $Myo7a^{Sh1/Sh1}$, 16/7. Abbreviations: DV, driver voltage; MET, mechanoelectrical transduction; OHC, outer hair cell; P, postnatal day; $P_o$, open probability.

became more pronounced (Fig. 5*A–D*). Although the number of stereocilia in the OHC first row was comparable between the two genotypes, those in rows 2 and 3 were significantly reduced in P10 $Myo7a^{Sh1/Sh1}$ mice compared with control animals ($P < 0.0001$, two-way ANOVA; Fig. 5*C*). The remaining stereocilia in the second and third row of OHCs from $Myo7a^{Sh1/Sh1}$ P10 mice also exhibited a significant reduction in their height ($P < 0.0001$, two-way ANOVA; Fig. 5*D*). The substantial changes in the transducing stereocilia caused a large reduction in the MET current from OHCs of P10 $Myo7a^{Sh1/Sh1}$ mice compared with littermate controls (Fig. 6*A–C*). However, the resting $P_o$ of the MET channel was no longer significantly different between the two genotypes (Fig. 6*D*). Increasing the intracellular

concentration of BAPTA from 0.1 to 5 mM in P9 OHCs caused a significant increase in the resting $P_o$ of the MET channel in both genotypes (Fig. 6*E* and *F*), which contrasts with the findings from immature P5 OHCs (Fig. 3).

These results indicate that the absence of functional MYO7A in $Myo7a^{Sh1/Sh1}$ mice leads to the progressive shortening and loss of the transducing stereocilia, starting from around P6, which is mirrored by the reduced MET current. Surprisingly, the $P_o$ of the resting MET current, which was very small or absent in immature OHCs from $Myo7a^{Sh1/Sh1}$ mice (Fig. 3), became indistinguishable from that recorded in control mice after their onset of functional maturation (∼P7–P8 onwards; Fig. 6).

### Contribution of the lipid bilayer to the resting $P_o$ of the MET channel in $Myo7a^{Sh1/Sh1}$ OHCs

A previous study has indicated that the resting MET current in pre-hearing OHCs is regulated, in part, through

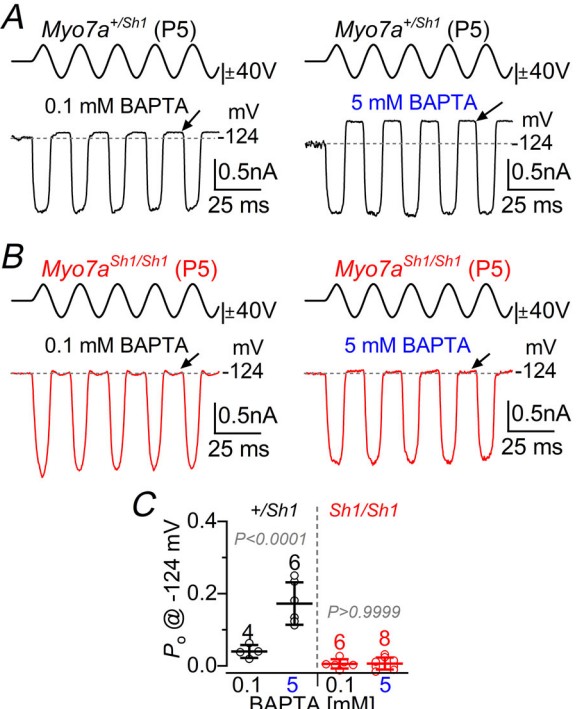

**Figure 3. Effect of intracellular BAPTA on the mechanoelectrical transduction currents in early postnatal outer hair cells from $Myo7a^{Sh1}$ mice**

*A* and *B*, saturating MET currents recorded from P5 OHCs of control $Myo7a^{+/Sh1}$ (*A*) and $Myo7a^{Sh1/Sh1}$ (*B*) mice at −124 mV in the presence of 0.1 mM (left) or 5 mM (right) intracellular BAPTA. The arrows indicate the closure of the transducer channel in response to inhibitory bundle stimuli. Note the increased resting MET current with 5 mM BAPTA in control OHCs but not in $Myo7a^{Sh1/Sh1}$ cells. *C*, resting $P_o$ of the MET channel recorded from OHCs of both genotypes using 0.1 or 5 mM intracellular BAPTA at −124 mV. Statistical comparisons between the genotypes are from one-way ANOVA, Tukey's *post hoc* test. Data in *C* are shown as the mean ± SD. Number of OHCs/mice: $Myo7a^{+/Sh1}$, 4/4 (0.1 mM BAPTA) and 6/4 (5 mM BAPTA); $Myo7a^{Sh1/Sh1}$, 6/6 (0.1 mM BAPTA) and 8/6 (5 mM BAPTA). Abbreviations: MET, mechanoelectrical transduction; OHC, outer hair cell; P, postnatal day; $P_o$, open probability.

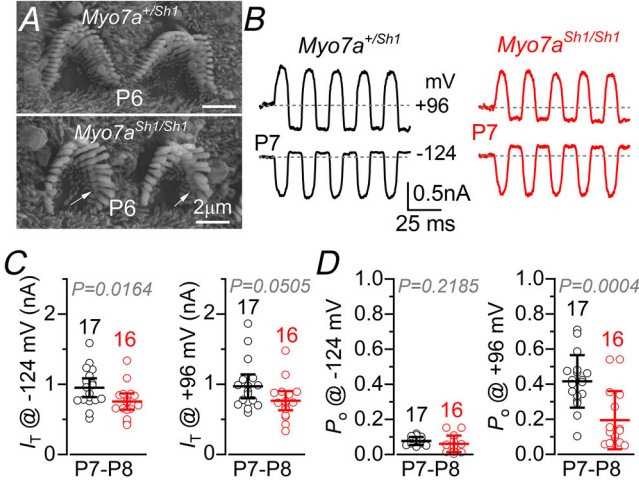

**Figure 4. Mechanoelectrical transduction currents from outer hair cells of $Myo7a^{Sh1}$ mice around the onset of their functional maturation**

*A*, scanning electron microscope images showing the hair bundles of apical-coil OHCs from control (top) and $Myo7a^{Sh1/Sh1}$ (bottom) mice at P6. Arrows point to a few missing stereocilia in the hair bundles of $Myo7a^{Sh1/Sh1}$ mice. *B*, saturating MET currents recorded from apical-coil OHCs of P7 control (left) and littermate (right) $Myo7a^{Sh1/Sh1}$ mice, i.e. around the onset of OHC functional maturation (P7–P8; Abe et al., 2007; Marcotti & Kros, 1999). Recordings are shown as described in Fig. 2. Note the resting MET current at both −124 and +96 mV in the OHC from a $Myo7a^{Sh1/Sh1}$ mouse. *C* and *D*, maximum current size (*C*) and resting $P_o$ of the MET channel (*D*) recorded from P7–P8 OHCs of both genotypes at −124 mV (left) and +96 mV (right). Statistical values were obtained using Student's *t* test. Data in *C* and *D* are shown as the mean ± SD. Number of OHCs/mice: $Myo7a^{Sh1}$, 17/8 and $Myo7a^{Sh1/Sh1}$, 16/9. Abbreviations: MET, mechanoelectrical transduction; OHC, outer hair cell; P, postnatal day; $P_o$, open probability.

the lipid bilayer around the MET channel, helping to set its $P_o$ within the range of greatest sensitivity (Peng et al., 2016). Therefore, we tested whether this mechanism could influence the resting $P_o$ of the MET channel in *Myo7a*$^{Sh1/Sh1}$ mice (Fig. 7). M$\beta$CD is a cyclic glucose heptamer that sequesters cholesterol and changes the fluidity and thickness of the lipid bilayer (Beurg et al., 2024). MET current recordings were performed from P5 mice as described in Fig. 2, but with the extracellular solution in the recording chamber and in the fluid jet containing different concentrations of M$\beta$CD (see Methods). Although compounds that alter the composition of the lipid bilayer are known to cause several secondary effects (e.g. reduction in the MET current; Beurg et al., 2024), we optimized their concentration and application time to limit these. Using this experimental approach, the MET current recorded from P5 OHCs in the presence of M$\beta$CD was similar not only between *Myo7a*$^{+/Sh1}$ and *Myo7a*$^{Sh1/Sh1}$ ($P = 0.3899$, two-way ANOVA; Fig. 7A–C), but also to that

recorded in the absence of M$\beta$CD ($P = 0.7967$, two-way ANOVA; Fig. 2C). Interestingly, we also found that M$\beta$CD reinstated the resting $P_o$ of the MET channel in OHCs from P5 *Myo7a*$^{Sh1/Sh1}$ mice to values comparable to those recorded in control mice at both negative and positive membrane potentials (Fig. 7A–E).

We also examined the potential contribution of PIP$_2$, another key membrane lipid enriched at the stereociliary tips of hair cells. PIP$_2$ has previously been shown to be essential for maintaining the normal amplitude and adaptation properties of the MET current (Cunningham et al., 2020; Effertz et al., 2017; Hirono et al., 2004). For these experiments, PIP$_2$ synthesis was prevented by using PAO in the extracellular solution (Fig. 7F–H). In the presence of extracellular PAO, both the maximum size of

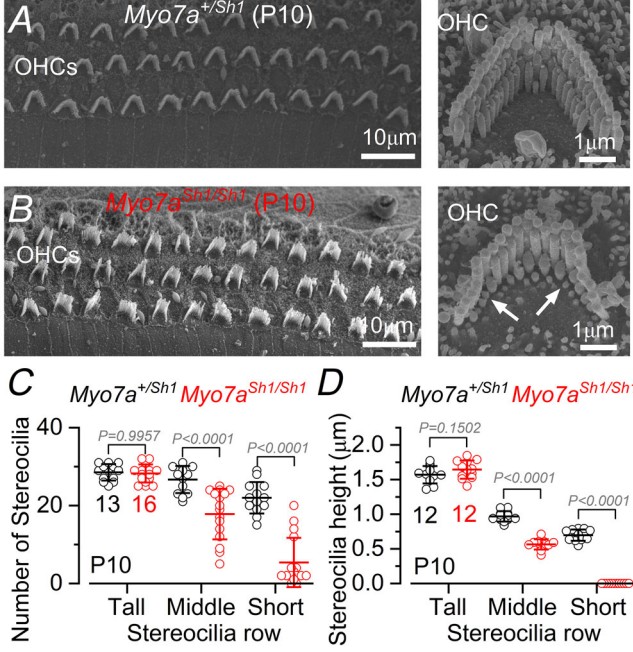

**Figure 5. Hair-bundle characteristics from outer hair cells of *Myo7a*$^{Sh1}$ mice after the onset of their functional maturation**
*A* and *B*, scanning electron microscope images showing the typical hair-bundle structure from OHCs of control (*A*) and *Myo7a*$^{Sh1/Sh1}$ (*B*) mice at P10. Right panels show an expanded view of a hair bundle for each genotype. Note that the hair bundle of the OHC from a *Myo7a*$^{Sh1/Sh1}$ mouse has lost several stereocilia and some have shortened (arrows). *C* and *D*, number (*C*) and height (*D*) of stereocilia in the tallest, middle and shortest rows in the hair bundles of OHCs from *Myo7a*$^{+/Sh1}$ and *Myo7a*$^{Sh1/Sh1}$ mice. Data are shown as the mean ± SD. Number of OHCs/mice: *Myo7a*$^{+/Sh1}$, 13/6 and *Myo7a*$^{Sh1/Sh1}$, 16/7. Statistical values are from two-way ANOVA, Šídák's multiple comparisons *post hoc* test. Abbreviations: OHC, outer hair cell; P, postnatal day.

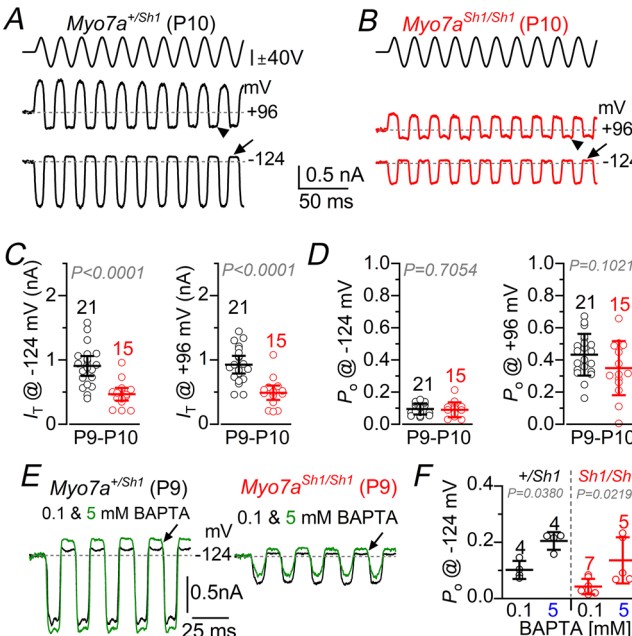

**Figure 6. Mechanoelectrical transduction currents from outer hair cells of *Myo7a*$^{Sh1}$ mice after the onset of their functional maturation**
*A* and *B*, saturating MET currents recorded from apical-coil OHCs of P10 control (*A*) and littermate *Myo7a*$^{Sh1/Sh1}$ (*B*) mice. *C* and *D*, maximum current size (*C*) and resting $P_o$ (*D*) of the MET channel recorded from OHCs of P9–P10 control and *Myo7a*$^{Sh1/Sh1}$ mice at −124 mV (left) and +96 mV (right). *E*, saturating MET currents recorded from P9 OHCs of control (left) and *Myo7a*$^{Sh1/Sh1}$ (right) mice elicited at −124 mV in the presence of 0.1 mM (black traces) and 5 mM (green traces) intracellular BAPTA. At P9, the resting MET current increased with 5 mM BAPTA in both control and *Myo7a*$^{Sh1/Sh1}$ OHCs. *F*, resting $P_o$ of the MET channel in P9 OHCs from the two genotypes in the presence of the two different BAPTA concentrations. Data in *C*, *D* and *F* are plotted as the mean ± SD. Statistical tests in *C* and *D* were performed using Student's *t* test and in *F* with Tukey's multiple comparisons test, one-way ANOVA. Number of OHCs/mice: *Myo7a*$^{+/Sh1}$, 21/11 and *Myo7a*$^{Sh1/Sh1}$, 15/9. Abbreviations: MET, mechanoelectrical transduction; OHC, outer hair cell; P, postnatal day; $P_o$, open probability.

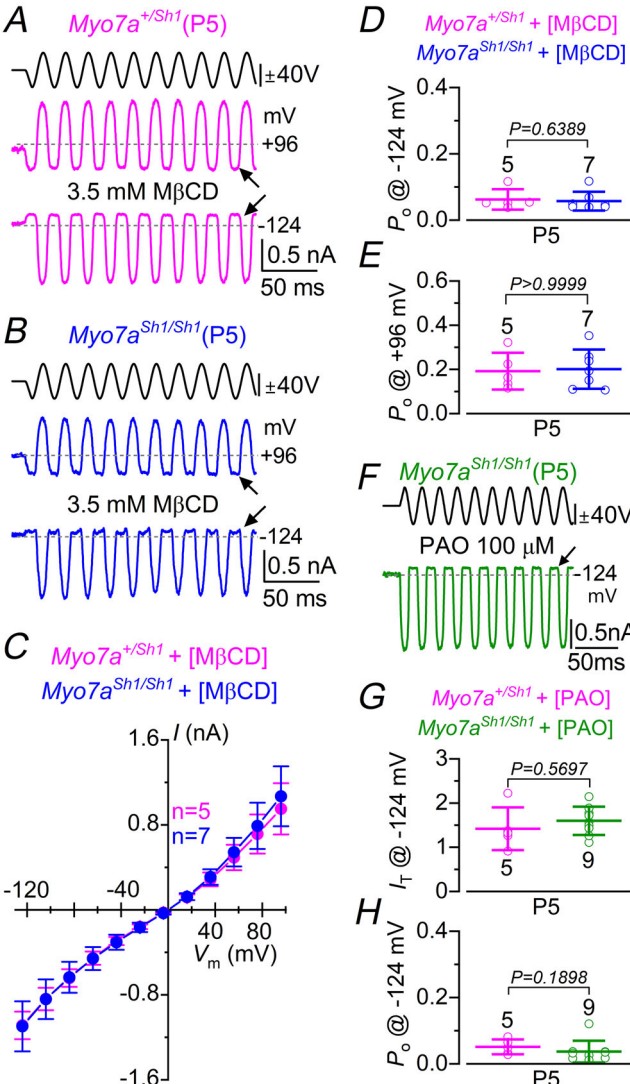

MβCD, methyl-β-cyclodextrin; MET, mechanoelectrical transduction; OHC, outer hair cell; P, postnatal day; PAO, phenylarsine oxide; $P_o$, open probability.

the MET current (Fig. 7*G*) and the resting $P_o$ of the MET channel (Fig. 7*H*) in P5 $Myo7a^{Sh1/Sh1}$ mice were not significantly different from those of $Myo7a^{+/Sh1}$ control OHCs.

These results indicate that in immature OHCs from $Myo7a^{Sh1/Sh1}$ mice, the resting MET current could be re-established in conditions that deplete some of the membrane lipid content.

### Mechanoelectrical transduction in developing IHCs from $Myo7a^{Sh1}$ mice

Considering the changes in MET currents recorded before and after the onset of functional maturation of OHCs, we tested whether developing IHCs from $Myo7a^{Sh1/Sh1}$ mice followed similar patterns. Different from OHCs, the normal maturation of IHCs begins a few days later, at around P12 (Kros et al., 1998).

The MET current in IHCs was elicited in the presence of 1 mM intracellular EGTA and at a membrane potential of −84 mV (Fig. 8*A*). As also shown in the immature P5 OHCs (Fig. 2), the resting $P_o$ of the MET channels in the IHCs was very small or absent in $Myo7a^{Sh1/Sh1}$ mice compared with control cells at P8–P9 (Fig. 8*B*). Although the size of the MET current elicited from immature P8–P9 IHCs of $Myo7a^{Sh1/Sh1}$ mice was comparable to that of control $Myo7a^{+/Sh1}$ cells (Fig. 8*A* and *C*), it was much more variable in the former. This was most probably attributable to the appearance of the initial morphological defects present in the hair bundle of P8–P9 IHCs of $Myo7a^{Sh1/Sh1}$ mice. By P10, both the number and height of the IHC stereocilia in rows 2 and 3 were significantly different between the two genotypes, especially in the latter row (Fig. 8*D–G*). Interestingly, row 2 exhibited a dysregulation in the stereocilia height, being on average taller than those in control cells (Fig. 8*E* and *G*). Similar to the effects observed in immature OHCs (Fig. 7), extracellular application of MβCD, which reduces membrane cholesterol content, restored the resting $P_o$ of the MET channels in IHCs of $Myo7a^{Sh1/Sh1}$ mice to levels comparable to those recorded in control cells of $Myo7a^{+/Sh1}$ mice (Fig. 8*H–J*). The progressive loss of transducing stereocilia in IHCs of $Myo7a^{Sh1/Sh1}$ mice resulted in a marked reduction in the MET current at, or shortly after, the onset of IHC maturation (P12–P13) compared with control cells (Fig. 9*A–C*). Nevertheless, similar to OHCs after their maturation onset (P9–P10; Fig. 6), the resting $P_o$ of the MET current in P12–P13 IHCs became comparable between the two genotypes (Fig. 9*D*).

**Figure 7. Effect of lipid removal on the mechanoelectrical transduction currents from outer hair cells of *shaker-1* mice**
*A* and *B*, MET currents recorded from OHCs of P5 control (*A*) and littermate $Myo7a^{Sh1/Sh1}$ (*B*) mice in the presence of 3.5 mM MβCD in the extracellular solutions surrounding the hair bundles. Recordings were obtained as described in Fig. 2. Note that the OHC from $Myo7a^{Sh1/Sh1}$ exhibits a resting MET current at both −124 and +96 mV, which was not present in age-matched OHCs recorded without MβCD (Fig. 2). *C*, average peak-to-peak MET current as a function of the OHC membrane potential from P5 control and $Myo7a^{Sh1/Sh1}$ mice with or without MβCD. The size of the MET current was not significantly different between the two experimental conditions (*P* = 0.3899, two-way ANOVA). *D* and *E*, resting $P_o$ of the MET channel obtained from P5 OHCs at −124 mV (*D*) and +96 mV (*E*) in control and $Myo7a^{Sh1/Sh1}$ mice. Number of OHCs/mice: $Myo7a^{+/Sh1}$, 5/4 and $Myo7a^{Sh1/Sh1}$, 7/3. *F*, example of MET current recorded from a P5 OHC in the presence of 100 μM extracellular PAO at −124 mV. *G* and *H*, average maximum size of the MET current (*G*) and resting $P_o$ of the MET channel (*H*) obtained from P5 OHCs at −124 mV in the different experimental conditions. Statistical comparisons in *D*, *E*, *G* and *H* are from the Mann–Whitney *U* test. All average data are plotted as the mean ± SD. Number of OHCs/mice: $Myo7a^{+/Sh1}$, 5/5 and $Myo7a^{Sh1/Sh1}$, 9/5. Abbreviations:

## Hair-bundle stiffness in pre- and post-hearing IHCs

The above findings indicate that reducing membrane lipid content in both OHCs and IHCs facilitates the opening of the MET channels, even in the absence of functional MYO7A. This effect could arise from either a decrease in the bundle stiffness or a reduction in the mechanical force required for gating the channel.

Because bundle stiffness is typically assessed using force-step stimuli rather than sinusoidal stimulation, we initially conducted experiments in which saturating MET currents were induced by step displacements of the hair bundle, while simultaneously monitoring their position using a fast camera (see Methods). As shown in Fig. 8, the resting MET current was either very small or absent in immature IHCs of $Myo7a^{Sh1/Sh1}$ mice, which was evident by the rightward shift of the current–displacement transfer function compared with control cells (Fig. 10*A*). Application of the cholesterol-sequestering M$\beta$CD

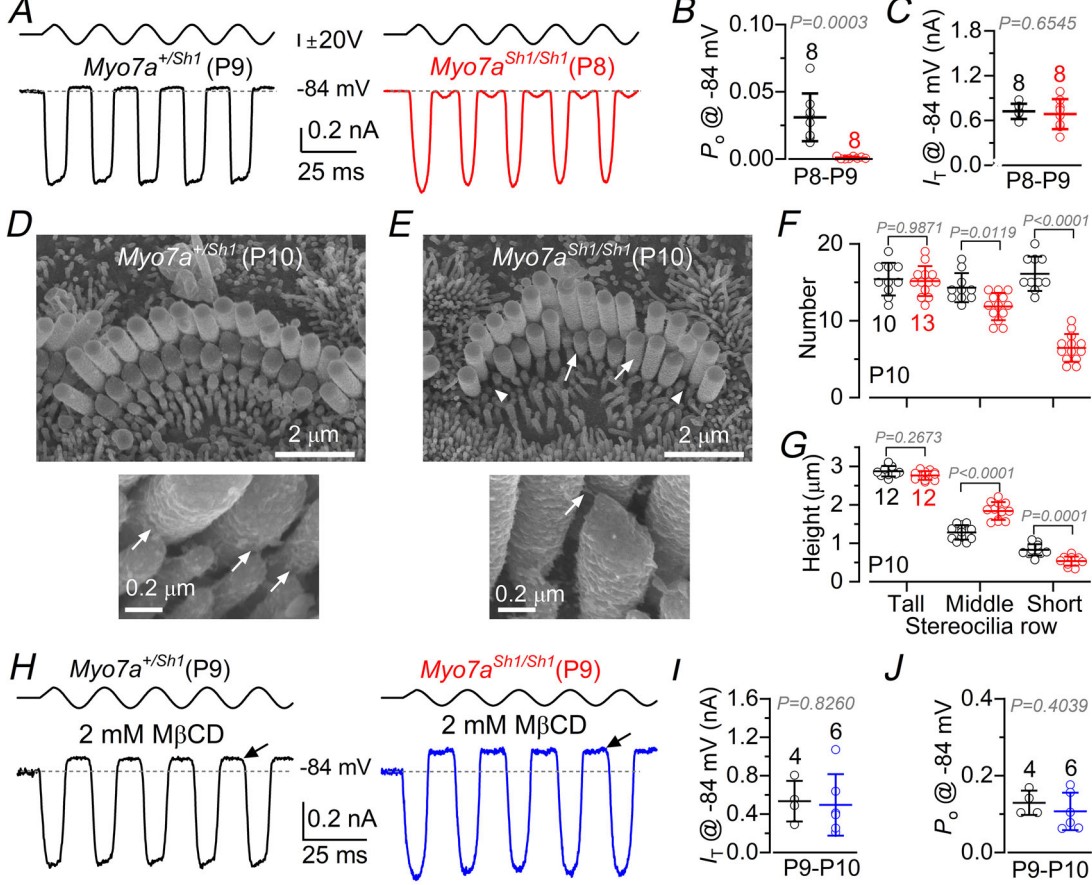

**Figure 8. Mechanoelectrical transduction from immature inner hair cells of *shaker-1* mice**

*A*, saturating MET currents recorded from apical-coil immature IHCs of P9 control (left) and P8 $Myo7a^{Sh1/Sh1}$ (right) mice at a membrane potential of −84 mV in the presence of 1 mM intracellular EGTA. *B* and *C*, average resting $P_o$ of the MET channel (*B*) and maximum current size (*C*) recorded from P8–P9 IHCs of both genotypes at −84 mV. Statistical comparisons were made with Student's *t* test. *D* and *E*, examples of scanning electron microscope images showing the hair-bundle structure from apical coil IHCs of control (*D*) and $Myo7a^{Sh1/Sh1}$ (*E*) P10 mice. Note the uneven height of the middle row of stereocilia (arrows) and their occasional absence (arrowheads) in IHCs of $Myo7a^{Sh1/Sh1}$ mice. Lower panels show expanded views of a hair bundle for each genotype, highlighting the presence of tip links (arrows) in both. *F* and *G*, number (*F*) and height (*G*) of stereocilia in the tallest, middle and shortest rows in the hair bundles of IHCs from $Myo7a^{+/Sh1}$ and $Myo7a^{Sh1/Sh1}$ mice. All average data are plotted as the mean ± SD. Number of IHCs/mice: $Myo7a^{+/Sh1}$, 10/4 and $Myo7a^{Sh1/Sh1}$, 13/5. Statistical values shown are from two-way ANOVA, with Šídák's multiple comparisons *post hoc* test. *H*, MET currents recorded from P9 IHCs of control (left) and littermate $Myo7a^{Sh1/Sh1}$ (right) mice measured at −84 mV and in the presence of 2 mM M$\beta$CD in the extracellular solution. Recordings were obtained as described in Fig. 2. Note that the resting MET current is present in the IHC from a P9 $Myo7a^{Sh1/Sh1}$ mouse. *I* and *J*, average MET current (*I*) and resting $P_o$ of the MET channel (*J*) obtained from P9–P10 IHCs at −84 mV in control and littermate $Myo7a^{Sh1/Sh1}$ mice. Statistical comparisons were performed using Student's *t* test. All average data are plotted as the mean ± SD. Number of IHCs/mice: $Myo7a^{+/Sh1}$, 4/3 and $Myo7a^{Sh1/Sh1}$, 6/4. Abbreviations: IHC, inner hair cell; M$\beta$CD, methyl-$\beta$-cyclodextrin; MET, mechanoelectrical transduction; P, postnatal day; $P_o$, open probability.

(2 mM) restored the resting MET current in the IHCs of *Myo7a$^{Sh1/Sh1}$* mice (Fig. 10*A*). We then assessed whether the absence of functional MYO7A or changes in membrane cholesterol affected bundle mechanics. For these additional experiments, we calculated hair-bundle stiffness by displacing the stereocilia using smaller, constant force-step stimuli that produced displacements of up to about ±120 nm (Fig. 10*A*, right panel), which elicited MET channel gating in the three experimental conditions (Fig. 10*B*). At P10, the same step stimuli evoked comparable bundle displacements in immature IHCs from *Myo7a$^{+/Sh1}$*, *Myo7a$^{Sh1/Sh1}$* and *Myo7a$^{Sh1/Sh1}$* cells treated with M$\beta$CD (Fig. 10*B* and *C*). The apparent steady-state bundle stiffness within the 120 nm displacement range was also similar among the three conditions (Fig. 10*C*), arguing against a possible direct role of MYO7A in setting the tip-link resting tension, because the gating stiffness has been shown to contribute ~20% of the total stiffness of IHC bundles (Tobin et al., 2019). Following the onset of maturation (P13), identical fluid-jet stimuli produced significantly larger bundle displacements and a marked reduction in stiffness in both control and *Myo7a$^{Sh1/Sh1}$* mice (*P* < 0.0001 for all comparisons between P10 and P13, one-way ANOVA, Tukey's *post hoc* test; Fig. 10*D* and *E*). However, because of the considerable variability observed in the IHCs of P13 *Myo7a$^{Sh1/Sh1}$* mice, differences in bundle displacement (Fig. 10*E*, left) and stiffness (Fig. 10*E*, right) relative to control cells were small or not statistically significant. This variability is likely to reflect the heterogeneity in IHC

bundle morphology in *Myo7a$^{Sh1/Sh1}$* mice, which ranged from nearly normal to bundles showing severe loss of the third row of stereocilia.

These results indicate that in immature IHCs, the absence of functional MYO7A leads to a reduced or absent resting MET current. However, lipid depletion is likely to reduce the force required to gate the MET channel, allowing hair cells to regain their resting $P_o$ even in the absence of functional MYO7A.

## Transcriptional changes in the *Myo7a$^{Sh1}$* mouse

To gain a better understanding of how the missense mutation in *Myo7a$^{Sh1}$* affected cochlear development, we performed RNA-sequencing from control *Myo7a$^{+/Sh1}$* and *Myo7a$^{Sh1/Sh1}$* P10 mice. After genotyping, both cochleae from four or five *Myo7a$^{+/Sh1}$* or *Myo7a$^{Sh1/Sh1}$* mice were pooled for RNA extraction. Principal component analysis of the top variable genes in each sample showed a clear separation between the *Myo7a$^{+/Sh1}$* and *Myo7a$^{Sh1/Sh1}$* mice, with principal component 1 (78% of the variance explained) dividing the samples by genotype. We next used DeSEQ2 (Love et al., 2014) to perform differential expression analysis of the two different genotypes and found that the missense mutation in *Myo7a$^{Sh1}$* does not significantly alter the expression levels of the *Myo7a* RNA (*P* = 0.2339, Student's unpaired *t*-test). This result also agrees with the expression level being virtually normal in these mice (Mburu et al., 1997). Although multiple splice isoforms of *Myo7a* have been described in the cochlea (Li et al., 2020), our dataset only revealed the presence of the canonical long isoforms, and no difference in splicing was found in *Myo7a$^{Sh1/Sh1}$* mice. Collectively, this suggests that the transcriptional changes observed in *shaker-1* mutant mice are attributable to the missense mutation in *Myo7a$^{Sh1}$* and not to changes in *Myo7a* expression.

Differential expression analysis found 564 genes that were downregulated and 37 genes that were upregulated in the cochlea of *Myo7a$^{Sh1/Sh1}$* compared with control *Myo7a$^{+/Sh1}$* mice (Supplemental Table 1; Fig. 11). Pathway analysis on the downregulated genes showed an enrichment for genes related to neuronal development and chemical transcription across synapses (Fig. 11*A*), consistent with previous morphological observations indicating changes in the cochlear innervation (Kikuchi & Hilding, 1965; Shnerson et al., 1983). Among the most downregulated genes (Fig. 11*B*), we found *Mafa*, which is expressed in type I spiral ganglion neurons (Jean et al., 2023) and *Dnajc5b* expressed in hair cells (Liu et al., 2014). Notably, we also observed decreased expression of several voltage-gated K$^+$ channels, Ca$^{2+}$-binding proteins and Atp proteins (Fig. 11*B*; Supplemental Table 1). Additionally, we found a decreased expression in *Calb2*, which is present in both the hair cells and spiral

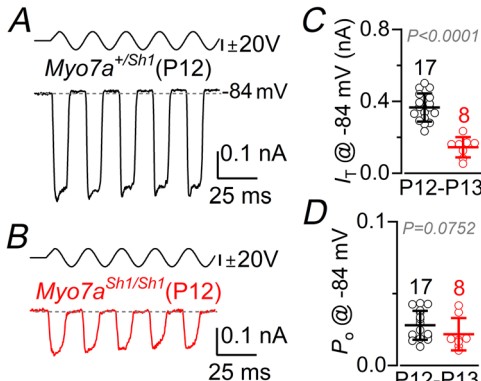

**Figure 9. Mechanoelectrical transduction in inner hair cells from *shaker-1* mice following their onset of maturation**
*A* and *B*, saturating MET currents recorded from IHCs of P12 control (*A*) and littermate *Myo7a$^{Sh1/Sh1}$* (*B*) mice. *C*, maximum size of the MET current recorded from P12–P13 IHCs of both genotypes at −84 mV. *D*, resting $P_o$ of the MET channel in P12–P13 IHCs from the two genotypes. Data in *C* and *D* are plotted as the mean ± SD. Statistical tests were performed using the Mann–Whitney *U* test. Number of IHCs/mice: *Myo7a$^{+/Sh1}$*, 17/8 and *Myo7a$^{Sh1/Sh1}$*, 8/3. Abbreviations: IHC, inner hair cell; MET, mechanoelectrical transduction; P, postnatal day; $P_o$, open probability.

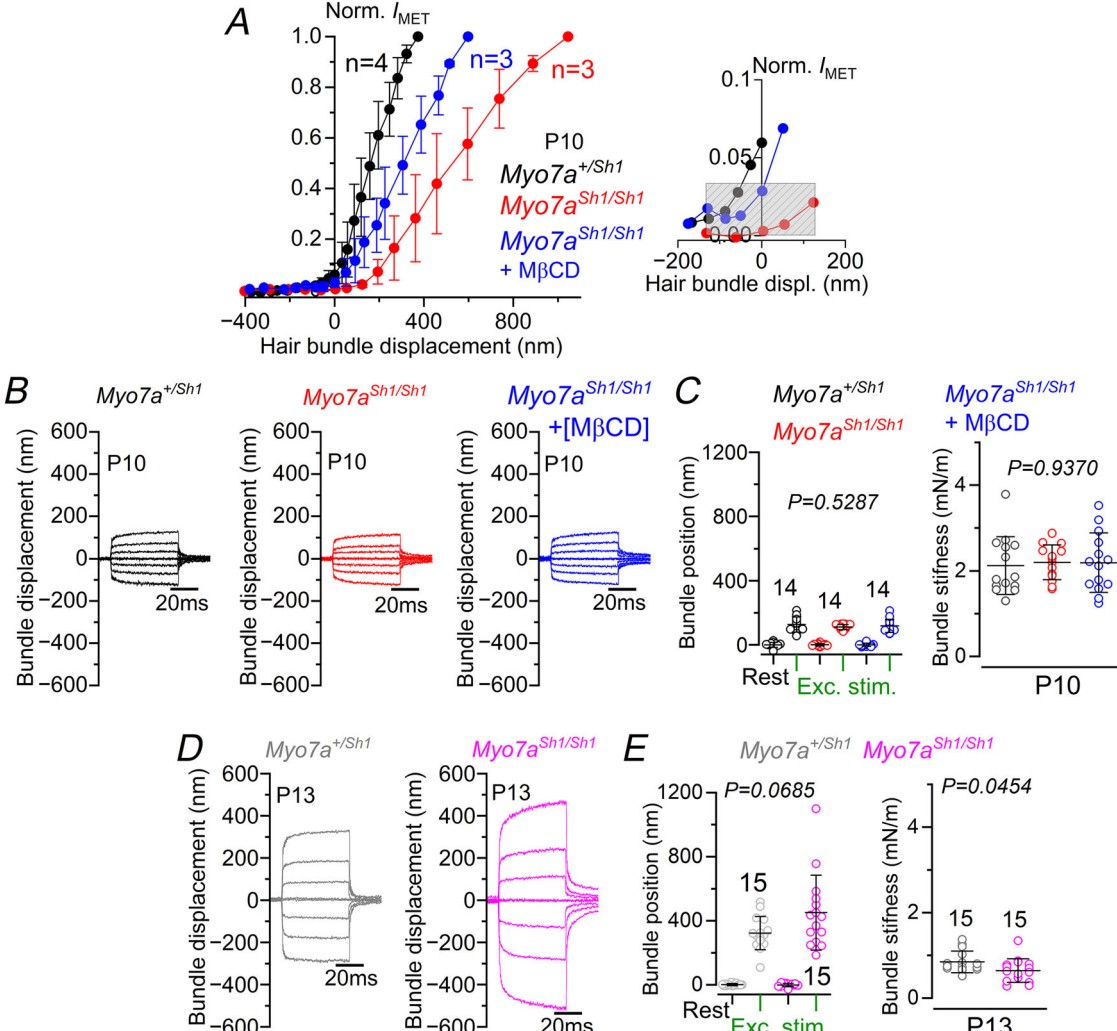

**Figure 10. Changes in hair-bundle stiffness in maturing inner hair cells from *shaker-1* mice**
*A*, normalized peak MET current as a function of hair-bundle displacement from P10 IHCs of control (*Myo7a*$^{+/Sh1}$),
*Myo7a*$^{Sh1/Sh1}$ and *Myo7a*$^{Sh1/Sh1}$ + M$\beta$CD mice. MET currents were recorded at the holding potential of −84 mV.
For these recordings, hair bundles were displaced by ±10 V (*Myo7a*$^{+/Sh1}$), ±20 V (*Myo7a*$^{Sh1/Sh1}$ + M$\beta$CD) and ±30
V (*Myo7a*$^{Sh1/Sh1}$) from a constant distance between the bundle and the fluid jet of 15 μm. Note the leftward shift
of the current–displacement curve in *Myo7a*$^{Sh1/Sh1}$ mice, which returned to near control values in the presence of
2 mM M$\beta$CD. The right panel shows a magnified view of the bundle displacement region around the activation
of the MET channel (grey area), which was used to measure the bundle stiffness in the following panels. Data are
shown as the mean ± SD. Number of IHCs/mice: *Myo7a*$^{+/Sh1}$, 4/3; *Myo7a*$^{Sh1/Sh1}$, 3/3; and *Myo7a*$^{Sh1/Sh1}$ + M$\beta$CD,
3/2. *B*, average hair-bundle displacements from 14 IHCs each (average movement from all visible stereocilia) of P10
control (left), *Myo7a*$^{Sh1/Sh1}$ (middle) and *Myo7a*$^{Sh1/Sh1}$ + M$\beta$CD (2 mM; right) mice elicited by applying incremental
±0.5 V force-step stimuli up to ±5V. For clarity, only hair-bundle displacements to 0, ±1.5, ±3 and ±5 V are
shown. *C*, left panel, average steady-state bundle displacement at rest (Rest) and at the excitatory position (±5 V,
Exc. Stim.) from each of the 14 IHCs from P10 control, *Myo7a*$^{+/Sh1}$ and *Myo7a*$^{+/Sh1}$ + M$\beta$CD mice. Right panel,
stiffness of each individual IHC hair bundle in the different experimental conditions. Data are shown as the mean
± SD. Number of IHCs/mice: *Myo7a*$^{+/Sh1}$, 14/3; *Myo7a*$^{Sh1/Sh1}$, 14/3; and *Myo7a*$^{Sh1/Sh1}$ + M$\beta$CD, 14/3. *D*, average
hair-bundle displacement from 15 IHCs of post-hearing P13 control (left) and *Myo7a*$^{Sh1/Sh1}$ (right) mice elicited by
applying incremental ±0.5 V force-step stimuli up to ±5 V. *E*, left panel, average steady-state bundle position at
rest (Rest) and during excitation (±5 V, Exc. Stim.) from each of the 15 IHCs from P13 control and *Myo7a*$^{Sh1/Sh1}$
mice. Right panel, stiffness of each individual IHC hair bundle in the different experimental conditions. Statistical
analysis was performed using either one-way ANOVA (*C*) or Student's unpaired *t*-test (*E*). Data are shown as the
mean ± SD. Number of IHCs/mice: *Myo7a*$^{+/Sh1}$, 15/3 and *Myo7a*$^{Sh1/Sh1}$, 15/3. Abbreviations: IHC, inner hair cell;
M$\beta$CD, methyl-$\beta$-cyclodextrin; MET, mechanoelectrical transduction; P, postnatal day; $P_o$, open probability.

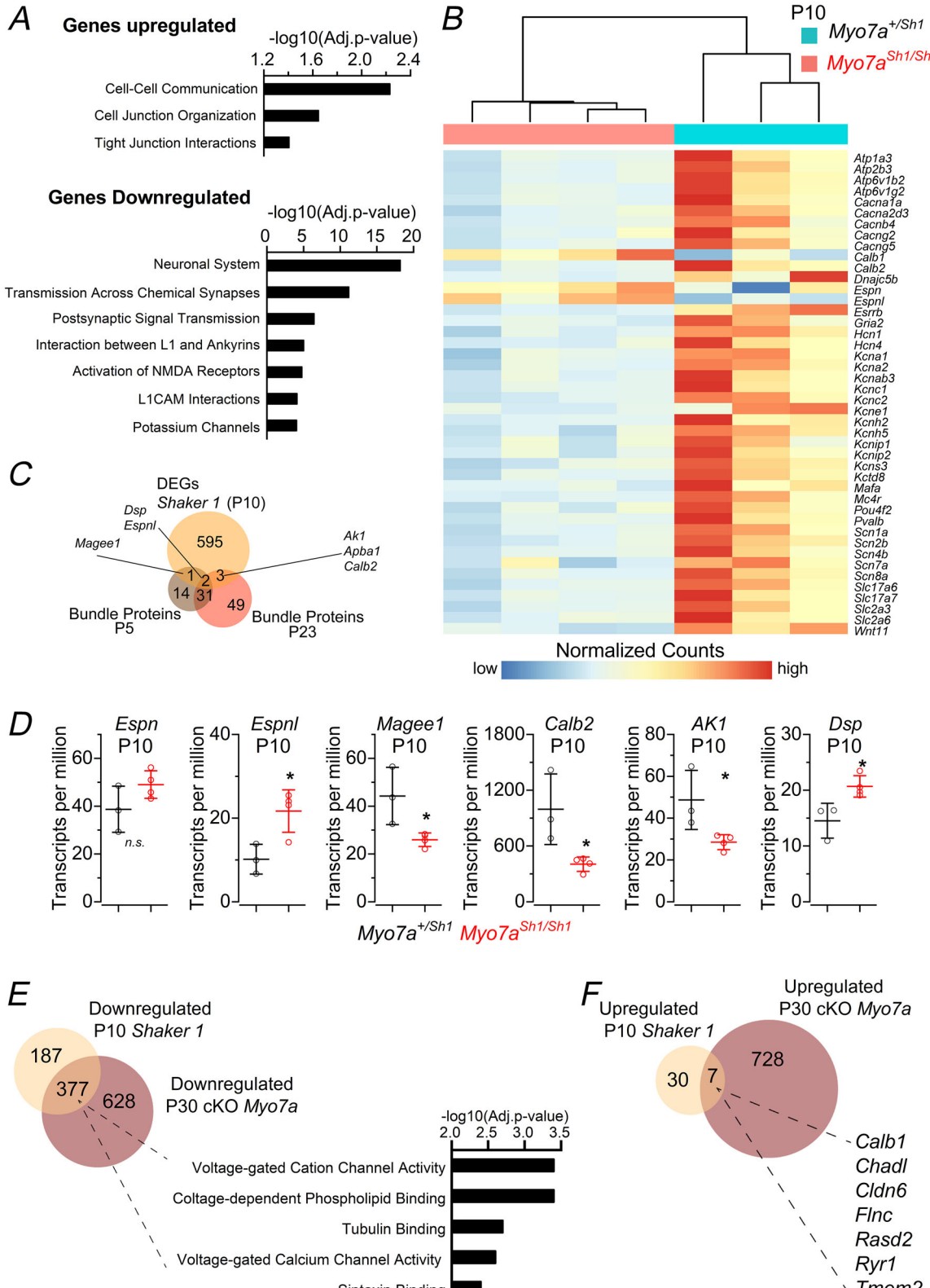

**Figure 11. Differentially expressed genes in *Myo7a^Sh1^* mice**

*A*, results of the pathways enrichment analysis associated with the upregulated (top) and downregulated (bottom) genes in *Myo7a^Sh1/Sh1^* mice. Enrichment analysis was performed using Enrichr. *B*, heatmap of differentially

expressed genes between control $Myo7a^{+/Sh1}$ and littermate $Myo7a^{Sh1/Sh1}$ mice at P10. Genes associated with the terms neuronal development and chemical transcription across synapses are highlighted. Unsupervised hierarchical clustering of the samples is visualized by the dendrogram on top of the heatmap, where the height of each line shows the similarity between samples or groups. *C*, Venn diagram of overlap between genes encoding for hair-bundle proteins identified at P5 and P23 (Ebrahim et al., 2016) and the genes whose expression changes in the $Myo7a^{Sh1/Sh1}$ mice at P10. *D*, transcripts per million from RNA-sequencing libraries for six of the identified genes that are downregulated or upregulated in $Myo7a^{Sh1/Sh1}$ mice at P10 ($Myo7a^{+/Sh1}$, three samples; $Myo7a^{Sh1/Sh1}$, four samples; each sample contained both cochleae from four or five mice). Statistical values were obtained from the *P*-value-adjusted Deseq2 differential analysis (Log2 fold change > 0.5, lfcse < 0.5, *P*-adjusted value < 0.01). *Espn*, $P = 0.1339$; *Espnl*, $P = 0.0207$; *Magee1*, $P = 0.0288$; *Calb2*, $P = 0.0261$; *AK1*, $P = 0.0371$; and *Dsp*, $P = 0.0226$. *E*, overlap between the genes downregulated in P30 *Myo7a* cKO mice (Underhill et al., 2025) and those downregulated in the $Myo7a^{Sh1/Sh1}$ mice. GO term analysis is provided for the 377 overlapping shared genes that are grouped in different functional categories (from top to bottom): GO:0022843; GO:0005544; GO:00155631; GO:0005245; GO:00199905. *F*, overlap between the genes upregulated in P30 *Myo7a* cKO mice (Underhill et al., 2025) and those regulated in the $Myo7a^{Sh1/Sh1}$ mice. All seven upregulated genes are listed. Abbreviations: cKO, conditional knockout; P, postnatal day.

ganglion neurons. Collectively, these data suggest that the missense mutation in $Myo7a^{Sh1}$, most likely due to its effects on the MET apparatus, is affecting the normal development of the spiral ganglion neurons.

Given that MYO7A is highly expressed in the stereociliary hair bundles (Morgan et al., 2016; Underhill et al., 2025), we investigated whether the $Myo7a^{Sh1}$ mutation was affecting the expression of any of its interacting partners or, more generally, proteins expressed in the stereocilia. MYO7A and the adaptor proteins USH1C (harmonin) and USH1G (sans) are believed to form a tripartite complex at the upper tip-link density (UTLD), which is localized at the insertion of the tip links to the side of the taller stereocilia (Adato et al., 2005; Boëda et al., 2002; Grati & Kachar, 2011; Yu et al., 2017). However, the localization of harmonin at the UTLD is independent from MYO7A, because it is normally expressed in *shaker-1* and *Myo7a* knockout mice (Grati & Kachar, 2011; Underhill et al., 2025). To determine the gene list for comparison, we pulled the hits from a published proteomics characterization of mouse hair bundles at P5 and P23 (Ebrahim et al., 2016). Given that our experiment was done at P10, we tested for shifts in bundle-associated and bundle-enriched proteins against both datasets (Fig. 11*C*). Collectively, few of the hair-bundle-associated genes were differentially expressed in $Myo7a^{Sh1/Sh1}$ mice, as might be expected from a structural mutation that may or may not lead to feedback regulation. However, two overlapping genes, *Espnl* and *Dsp*, were upregulated in the homozygous $Myo7a^{Sh1/Sh1}$, suggesting a possible compensatory mechanism aimed at maintaining hair-bundle integrity (Fig. 11*D*). Of these, only *Espnl* is expressed exclusively in hair cells, whereas the other overlapping genes were also expressed in spiral ganglion neurons or other cell types in the cochlea. ESPNL is highly enriched in the stereociliary bundles and has been found to be crucial for stereociliogenesis (Ebrahim et al., 2016). ESPNL has been shown to be concentrated primarily at the tips of the shorter second and third rows of stereocilia, which are those most affected

in *shaker-1* mice. Interestingly, we found that despite the upregulation of *Espnl* in $Myo7a^{Sh1/Sh1}$ mice (Fig. 11*D*), ESPNL puncta were not only significantly reduced but also mis-localized to the tip of the first row of stereocilia in both P9 IHCs and OHCs compared with controls ($P = 0.0002$, Student's unpaired *t*-test; Fig. 12*A*–*C*). It is, however, unlikely that ESPNL transport to the tips of the stereocilia of $Myo7a^{Sh1/Sh1}$ mice was impaired, because no immunostaining was detected in any other cellular membrane or intracellular compartments. Additionally, this was not attributable to lack of stereocilia in the hair cells of $Myo7a^{Sh1/Sh1}$ mice (e.g. P10; Fig. 8*D*–*G*), which also showed normal labelling of other key proteins required for MET current, such as PCDH15 and PMCA (Fig. 12*D* and *E*). Using the STRING database, we found no change in expression of the EPSNL interacting proteins. The discrepancy between RNA (*Espnl* upregulation) and protein (EPSNL downregulation) levels is not surprising considering that a recent study has reported <0.5 concordance between the two values (Wegler et al., 2019).

We also compared our findings with our previously published (Underhill et al., 2025) model for the conditional *Myo7a* deletion at P30, where the predominant transcriptional changes were found in those mice. A large subset of the downregulated genes in the $Myo7a^{Sh1/Sh1}$ mice (∼66%) were also downregulated in the *Myo7a* conditional knockout mice at P30, indicating a common response pathway, with GO terms revealing changes in voltage-gated ion channels and tubulin binding (Fig. 11*E*). Of the modest number of upregulated genes, 19% of these were shared with the *Myo7a* conditional knockout mice (Fig. 11*F*).

## Hair-bundle disorganization in $Myo7a^{Sh1/Sh1}$ mice is exacerbated by noise exposure

We tested whether the integrity of the hair bundles lacking functional MYO7A was affected by noise insults (see

Methods), which has previously been demonstrated in adult *Myo7a* conditional knockout mice lacking MYO7A in the hair cells (Underhill et al., 2025). We exposed control and *Myo7a*$^{Sh1/Sh1}$ mice to a noise level that caused a temporary threshold shift in ABR thresholds. Considering the fast progressive hearing loss in *Myo7a*$^{Sh1/Sh1}$ mice, noise exposure was performed at P14, which is a couple of days after hearing onset. ABR thresholds to pure tones (3–30 kHz) were measured immediately after noise exposure, then again at P18 to assess the possible recovery from the temporary threshold shift (Fig. 13*A* and *B*). These ABR thresholds were compared with those recorded from age-matched unexposed mice. In *Myo7a*$^{+/Sh1}$ control mice, the elevated ABR thresholds after noise exposure at P14 returned to near-normal levels after 4 days (Fig. 13*A*). In contrast, the ABR thresholds from *Myo7a*$^{Sh1/Sh1}$ mice failed to recover (Fig. 13*B*), highlighting a higher sensitivity of mice lacking

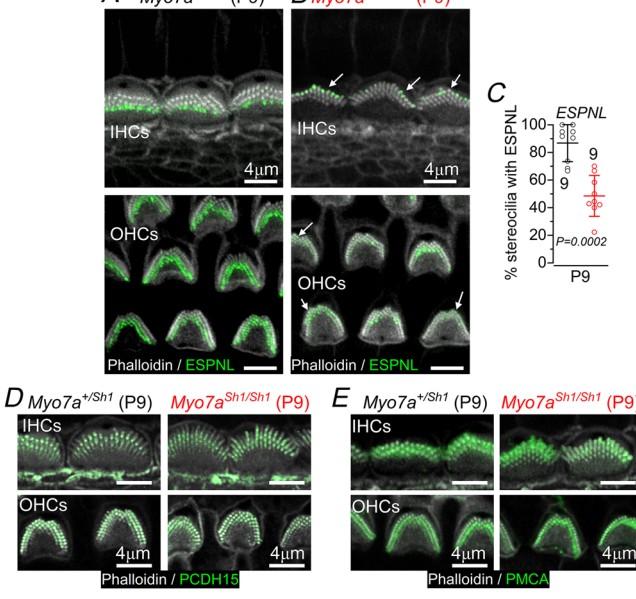

**Figure 12. ESPNL is strongly reduced in the stereocilia of hair cells of *shaker-1* mice**
*A* and *B*, ESPNL (green) localization in IHCs (top) and OHCs (bottom) at the tips of the stereocilia in P9 control (*A*) and *Myo7a*$^{Sh1/Sh1}$ (*B*) mice. Stereocilia are stained with phalloidin (grey). The control hair cells exhibit ESPNL puncta primarily at the tips of the two shortest rows of stereocilia, whereas in *Myo7a*$^{Sh1/Sh1}$ mice, these puncta were also mis-localized to the tips of the first row in both IHCs and OHCs. *C*, average of ESPNL puncta at OHC stereocilia, shown as percentage of total stereocilia. Data are plotted as the mean ± SD. Number of OHCs/mice: *Myo7a*$^{+/Sh1}$, 9/3 and *Myo7a*$^{Sh1/Sh1}$, 9/3. Statistical tests were performed using the Mann–Whitney *U* test. The number of OHCs measured is shown above or below the data. *D* and *E*, normal labelling of the stereocilia by PCDH15 (*D*) and PMCA (*E*) in both IHCs (top) and OHCs (bottom) of P9 control (left panels) and *Myo7a*$^{Sh1/Sh1}$ (right panels) mice (three mice per genotype for both *C* and *E*). Abbreviations: IHC, inner hair cell; OHC, outer hair cell; P, postnatal day.

functional MYO7A to noise exposure. The lack of ABR threshold recovery following noise exposure was most probably attributable to the increased disorganization of the stereociliary bundle in hair cells from mice lacking functional MYO7A, especially those of the IHCs (Fig. 13*C* and *D*).

## Discussion

Here, we show that a spontaneous missense mutation in the *Myo7a* gene (*Myo7a*$^{Sh1}$), which is known to interfere with the motor function of MYO7A (Gibson et al., 1995) and underlies Usher syndrome type 1B in humans (Weil et al., 1995), plays a crucial role in the maintenance of the morphological integrity of the stereociliary bundles in hair cells. Beginning from around the second postnatal week, the hair bundles of *Myo7a*$^{Sh1/Sh1}$ mice undergo progressive height dysregulation and loss of their second and third rows of transducing stereocilia, leading to deafness by ∼1 month of age. Noise exposure exacerbates the hair-bundle disruption and the progression of hearing loss in *Myo7a*$^{Sh1/Sh1}$ mice. In the absence of functional MYO7A, MET currents of immature OHCs and IHCs have normal maximal size but markedly reduced resting $P_o$ and $Ca^{2+}$ sensitivity. However, reduction of the cell membrane lipid content, which changes the fluidity or thickness of the membrane bilayer, rescues the resting MET current in immature hair cells from *Myo7a*$^{Sh1/Sh1}$ mice. Interestingly, from the onset of functional maturation, which occurs from around P6–P7 in OHCs (Marcotti & Kros, 1999) and P12 in IHCs (Kros et al., 1998), the MET channels show a normal resting $P_o$ and $Ca^{2+}$ sensitivity even in *Myo7a*$^{Sh1/Sh1}$ mice. The re-established resting $P_o$ in mature IHCs was accompanied by significantly reduced bundle stiffness compared with pre-hearing cells, thus decreasing the overall force required to gate the MET channels.

### MYO7A is required for the maintenance of the transducing stereocilia

The hair cells of the original *shaker-1* mutant mice (*Myo7a*$^{Sh1}$), which carry an arginine-to-proline missense mutation located in a poorly conserved surface loop of the motor head (Gibson et al., 1995, Mburu et al., 1997), exhibit normal expression of MYO7A, and their hair bundles develop the characteristic staircase structure (Shnerson et al., 1983; Self et al., 1998). However, the loss of MYO7A function has been shown to cause hair-bundle disorganization and substantial hearing loss by P15 (Mikaelian & Ruben, 1964; Self et al., 1998), followed by deafness and hair-cell degeneration at later stages (Lord & Gates, 1929; Mikaelian & Ruben, 1965; Shnerson et al., 1983). We found that OHCs from *Myo7a*$^{Sh1/Sh1}$ mice exhibit early signs of defective stereocilia from the end

of the first postnatal week, with some stereocilia in the second and third rows becoming shorter or longer. Given that elongation of the shortest rows of stereocilia normally stops at around P5 (Peng et al., 2009), our data fully support the current notion that the *Myo7a$^{sh1}$* mutation has little impact on the initial development of the hair bundles. However, by P10, the two shortest rows of stereocilia in both IHCs and OHCs from *Myo7a$^{Sh1/Sh1}$* mice are already severely affected, indicating that MYO7A is required primarily for stabilizing the transducing stereocilia, which contain the MET channels at their tip (Beurg et al., 2009). Although the predominant morphological defect in *Myo7a$^{Sh1/Sh1}$* mice was progressive loss of the transducing stereocilia, especially those in the shortest row, a more complex dysregulation was present in the second row, with some stereocilia growing significantly longer than those in control hair cells. Interestingly, MYO7A appears to have minimal or no impact on either the elongation or maintenance of the tallest row of stereocilia. A similar progressive loss of the transducing stereocilia has been reported in adult conditional *Myo7a* knockout mice (Underhill et al., 2025), indicating that MYO7A exerts a similar role in the hair bundles of both developing and adult hair cells.

Like other unconventional myosins in the cochlea (such as MYO3A and MYO15A), MYO7A uses its motor activity to traffic structural and actin-regulatory proteins to the stereocilia (Miyoshi et al., 2024; Moreland & Bird, 2022). Through the interaction with the cytoplasmic domain of PCDH15, MYO7A can traffic tip-link components within the hair bundle, a mechanism that is no longer evident in *shaker-1* mice (Senften et al., 2006). Another prominent role for MYO7A is to transport PDZD7–ADGRV1 to the base of the stereocilia, which is crucial for the formation of the ankle-link protein complex located in or immediately above the tapered region at the base of each stereocilium (e.g. Grati et al., 2012; McGee et al., 2006; Morgan et al., 2016). Indeed, the localization of ADGRV1 is disturbed in *shaker-1* mice (Michalski et al., 2007). However, ankle links are present only between P2 and P12 (Goodyear et al., 2005) and, as such, are unlikely to be the main or sole contributor to the hair-bundle phenotype observed in the hair cells of *Myo7a*-deficient mice during development (*Myo7a$^{Sh1/Sh1}$*) or in the adult cochlea (Underhill et al., 2025). The ESPN-1 paralogue ESPNL is another protein expressed in the stereocilia of developing hair cells and required for their normal staircase structure, although primarily in the hair cells located in the high-frequency

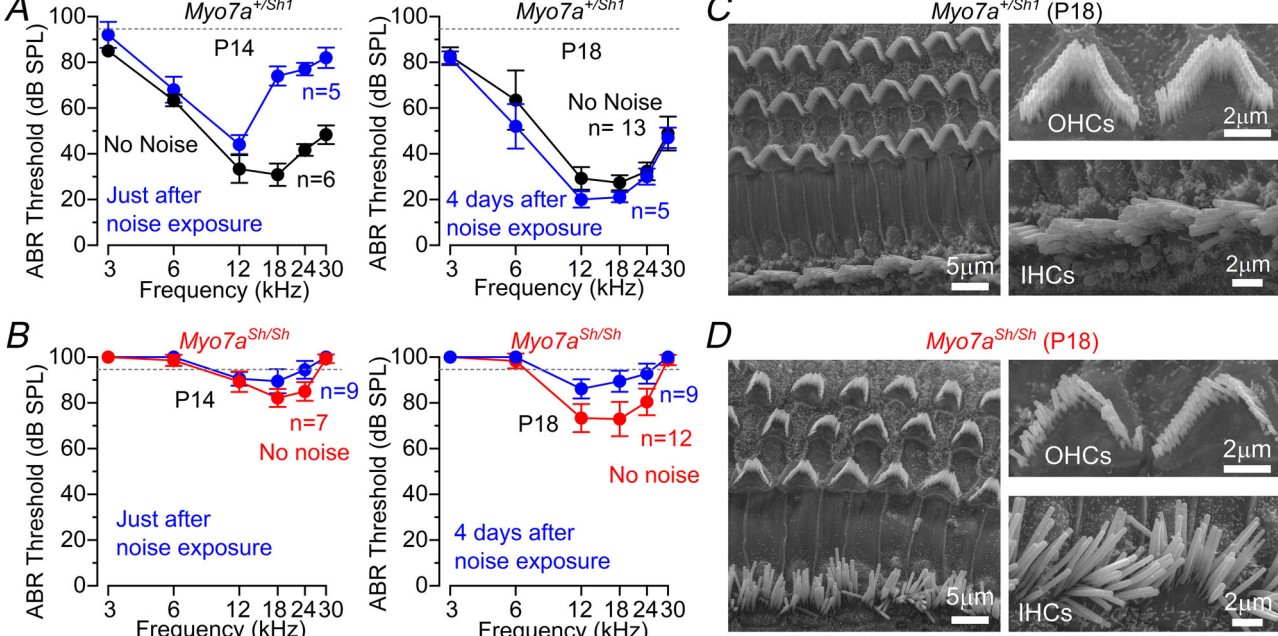

**Figure 13. Noise exposure exacerbates hearing loss of *Myo7a$^{Sh1/Sh1}$* mice**
*A* and *B*, average ABR thresholds for frequency-specific pure tone stimulation from 3 to 30 kHz recorded from *Myo7a$^{+/Sh1}$* (*A*, black) and littermate *Myo7a$^{Sh1/Sh1}$* (*B*, red) mice at P14 and P18. For mice exposed to noise (blue), recordings were performed at P14 (left panels; immediately after noise exposure) and at P18, 4 days after noise exposure (right panels). The ABR thresholds in control mice recovered almost completely by P18 after noise exposure, whereas those from *Myo7a$^{Sh1/Sh1}$* mice failed to do so. The number of mice tested for each age/strain is shown next to the average data. The dashed line represents the upper threshold limit of our system, 95 dB. *C* and *D*, scanning electron microscope images showing the OHC hair-bundle structure in the apical coil of the cochlea of P18 *Myo7a$^{+/Sh1}$* control mice (*C*) and *Myo7a$^{Sh1/Sh1}$* mice, which were both exposed to noise at P14 (*D*). Abbreviations: ABR, auditory brainstem response; IHC, inner hair cell; OHC, outer hair cell; P, postnatal day.

cochlear region (Ebrahim et al., 2016). By suppressing elongation of stereocilia, ESPNL has been suggested to contribute to the generation of the much shorter transducing stereocilia (Ebrahim et al., 2016). Likewise, we found that in the absence of functional MYO7A the height of the transducing stereocilia was strongly dysregulated, with some becoming taller and others too short. These abnormal stereocilia also exhibited a significantly reduced number of ESPNL-immunostained puncta. It is unlikely, however, that MYO7A is involved directly in transporting ESPNL to the stereocilia tip because this role has previously been ascribed to the myosin motors MYO3A and MYO3B (Ebrahim et al., 2016), the expression of which was not affected in *shaker-1* mice. Overall, the combined effect of several proteins that are directly or indirectly not correctly localized at the stereocilia compromises the morphological homeostasis of the hair bundles. Indeed, we found that mechanical stress, such as that experienced during noise exposure, exacerbates the progression of hair-bundle morphological dysfunction and hearing loss in *Myo7a*-deficient mice (see also Underhill et al., 2025).

### The resting MET current is not regulated solely by MYO7A

A recent functional study has indicated that hair cells seem to express multiple isoforms of MYO7A, with the canonical isoform *Myo7a-ΔC* primarily expressed in IHCs (Li et al., 2020). In pre-hearing *Myo7a-ΔC*-deficient mice (P7–P8), IHCs, but not OHCs, exhibited a greatly reduced resting $P_o$ of the MET channel (Li et al., 2020), supporting the idea that MYO7A regulates tip-link tension and thus the mechanical sensitivity of the MET channel. Interestingly, the findings by Li et al. (2020) are consistent with our results in age-matched *Myo7a^{Sh1/Sh1}* mice, where OHCs showed a normal resting $P_o$ after their onset of maturation, whereas in immature IHCs it was strongly reduced. These results also agree with recent findings showing that adult hair cells from *Myo7a*-deficient mice retain a normal resting $P_o$, at least until the MET current is fully abolished (Underhill et al., 2025). Altogether, these results strongly support the hypothesis that the gating of the MET channels might differ depending on the developmental stage of the hair cells. This could be attributable to developmental changes in either the transduction machinery and/or the membrane surrounding the MET channels. This hypothesis is plausible considering that the hair cells undergo several biophysical and morphological changes at their onset of maturation, including switching the MET channel pore-forming subunit from TMC2 to TMC1 (Pan et al., 2018) at ~P5–P6 in OHCs and by P13 in IHCs

(Beurg et al., 2018), ages that correspond to the different phenotypes observed in the hair cells from *shaker-1* mice.

The remaining question to address, therefore, is whether MYO7A is regulating the resting tension of the tip links by itself. This seems unlikely, because we have shown that the resting $P_o$ of the MET channels, which was absent in immature OHCs (P5) and IHCs (P9–P10) of *Myo7a^{Sh1/Sh1}* mice, was restored in IHCs following their onset of maturation or simply by changing the fluidity or thickness of the lipid bilayer in immature cells (Fig. 14). Considering that the size of the MET current in our experiments was unaffected by the application of either MβCD or PAO, the rescue of the MET channel resting $P_o$ could, at least in principle, be caused by either a decrease in bundle stiffness or a reduction in the mechanical force required for gating the channel. In

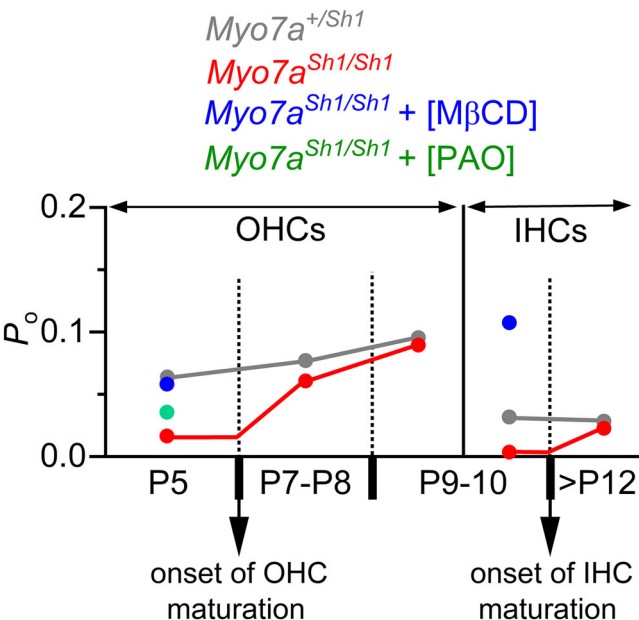

**Figure 14. Developmental changes in resting open probability of the mechanoelectrical transduction channel in the hair cells of *shaker-1* mice**

Summary of the developmental changes in MET channel resting $P_o$ in OHCs (measured at -124 mV) and IHCs (measured at -84 mV) between P5 and P13. Hair cell functional maturation begins around P6 in OHCs and at about P12 in IHCs, with P12 also marking the onset of hearing in mice. In *Myo7a^{Sh1/Sh1}* mice, the resting $P_o$ of the MET channel is markedly reduced or absent in immature OHCs and IHCs. However, after the onset of maturation in each cell type, resting $P_o$ increases to values comparable to those of control *Myo7a^{+/Sh1}* mice. Interestingly, reducing membrane lipid content either by depleting cholesterol with MβCD or by inhibiting PIP$_2$ synthesis with PAO, restores normal resting $P_o$ in both OHCs and IHCs, even in the absence of functional MYO7A. For simplicity, only average data are presented in this summary figure. Abbreviations: IHC, inner hair cell; MβCD, methyl-β-cyclodextrin; MET, mechanoelectrical transduction; OHC, outer hair cell; P, postnatal day; PAO, phenylarsine oxide; PIP$_2$, phosphatidylinositol-4,5-bisphosphate; $P_o$, open probability.

immature hair cells, bundle stiffness was unchanged in the absence of MYO7A or when the lipid content was depleted, indicating that the force transmission via the lipid bilayer to the MET channel has been changed, a mechanism that has been described extensively for other mechanosensitive channels, such as the bacterial MscL (Nomura et al., 2012). A force-conveying role of the lipid membrane to the gating of the MET channel has previously been proposed using a computational approach (Kim, 2015). Additionally, experiments from cochlear hair cells have shown that the toxin GsMTx4, a lipid-mediated modifier of cationic stretch-activated channels (Suchyna & Sachs, 2007) that acts by partitioning into the membrane and altering lipid packing (Suchyna et al., 2004), behaves as a gating modifier of MET channels, shifting its opening towards larger bundle displacements (Beurg et al., 2014; Peng et al., 2016). In this case, GsMTx4 has been proposed to increase the resting forces applied to the channel and thus the energy required to gate the MET channels, reducing its resting $P_o$ (Peng et al., 2016). This mechanism is further supported by the evidence that with maturation, the IHC bundles had significantly reduced their stiffness independently of the presence or absence of MYO7A, thus requiring less force to gate the MET channels and the restoration of its $P_o$ to normal level. Thus, the role of MYO7A in mechanoelectrical transduction could be that, during early stages of development, it is required directly or indirectly to deliver key proteins to the tip of the transducing stereocilia that normally facilitate the force transmitted through the lipid bilayer to the MET channels. In its absence, the force required to gate the channel is increased but overcome by affecting the lipid bilayer or by the significantly reduced hair-bundle stiffness in more mature hair bundles.

The finding that $PIP_2$ reduction in the membrane was able to rescue the resting $P_o$ of the MET channel, without affecting the overall size of the MET current, further supports the idea that the lipid bilayer is likely to mediate the force required to gate the MET channel. Previous studies have shown that $PIP_2$ binds to TMIE, a protein of the MET apparatus that interacts with the MET channel-forming subunits TMC1/TMC2, facilitating the gating of the channel, including the setting of the resting $P_o$ (Caprara et al., 2025; Cunningham et al., 2020; Effertz et al., 2017). A comparable dynamic has previously been identified in mechanosensitive TREK-1 channels, where membrane phospholipids, including $PIP_2$, regulate their gating (Chemin et al., 2005).

Overall, our results suggest that the gating of the MET channels might differ depending on the developmental stage of the hair cells. During hair cell development, the resting MET current appears to be regulated by the interaction of the MET channel with the lipid bilayer. Although MYO7A is contributing to setting the resting MET current

in immature hair cells, it is likely to do so via the transport of essential cargo proteins that interact directly or indirectly with the MET channel subunit components in the membrane.

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

## Additional information

### Data availability statement

The data that support the findings of this study are available from the corresponding authors. RNA-sequencing data have been deposited in GEO under accession number (GSE240605).

### Competing interests

The authors declare no conflict of interest.

### Author contributions

All authors helped with the collection and analysis of the data. All authors approved the final version of the manuscript and agree to be accountable for all aspects of the work in ensuring that questions related to the accuracy or integrity of any part of the work are appropriately investigated and resolved. All persons designated as authors qualify for authorship, and all those who qualify for authorship are listed.

### Funding

This work was supported by the RNID (G94) to W.M. and C.J.K., BBSRC (BB/T004991/1) and Wellcome Trust (224 326/Z/21/Z) to W.M.; BBSRC (BB/X000567/1) to S.L.J.; Wellcome Trust (300 350/Z/23/Z) to A.J.C. A.U. was supported by a PhD studentship from the MRC DiMeN Doctoral training Partnership to W.M.

For the purpose of Open Access, the author has applied a CC BY public copyright licence to any Author Accepted Manuscript version arising from this submission.

## Acknowledgements

We thank Michelle Bird for assistance with the mouse husbandry, and Catherine Gennery, Matthew Hool and Niovi Voulgari for their genotyping work.

## Keywords

cochlea, deafness, hair cell, ion channel, mechanoelectrical transduction, myosin motor

## Supporting information

Additional supporting information can be found online in the Supporting Information section at the end of the HTML view of the article. Supporting information files available:

**Peer Review History**
**Supplementary Table 1**

