## [Peer Review History · The Journal of Physiology]

Myosin 7a is required for maintaining the transducing stereocilia and for force transmission to the MET channel during cochlear hair cell development

Anna Underhill, Samuel Webb, Fiorella Carla Grandi, Adam James Carlton, Jing-Yi Jeng, Stuart Leigh Johnson, Corne J Kros, and Walter Marcotti
DOI: 10.1113/JP289623

Corresponding author(s): Walter Marcotti (w.marcotti@sheffield.ac.uk)

Review Timeline:

Submission Date:	27-Jun-2025
Editorial Decision:	06-Aug-2025
Revision Received:	03-Dec-2025
Editorial Decision:	06-Jan-2026
Revision Received:	13-Jan-2026
Accepted:	22-Jan-2026

Senior Editor: Kim Barrett

Reviewing Editor: Pawel Ferdek

Transaction Report:

Dear Dr Marcotti,

Re: JP-RP-2025-289623 "**The motor protein Myosin 7a regulates the maintenance of the two shortest rows of stereocilia in cochlear hair cells**" by Anna Underhill, Samuel Webb, Fiorella Carla Grandi, Adam Carlton, Stuart Leigh Johnson, Corne J Kros, and Walter Marcotti

Thank you for submitting your manuscript to The Journal of Physiology. It has been assessed by a Reviewing Editor and by 2 expert referees and we are pleased to tell you that it is potentially acceptable for publication following satisfactory major revision.

REVISION CHECKLIST:

We look forward to receiving your revised submission.

Yours sincerely,

Kim Barrett
Senior Editor
The Journal of Physiology

REQUIRED ITEMS

1) - Author photo and profile. First or joint first authors are asked to provide a short biography (no more than 100 words for one author or 150 words in total for joint first authors) and a portrait photograph. These should be uploaded and clearly labelled together in a Word document with the revised version of the manuscript. See Information for Authors for further details.

2) - You must start the Methods section with a paragraph headed Ethical approval (https://jp.msubmit.net/cgi-bin/main.plex?form_type=display_requirements#methods).

Research must comply with The Journal's policies regarding animal experiments (<https://physoc.onlinelibrary.wiley.com/hub/animal-experiments>) and adherence to these policies must be stated in the manuscript.

Authors should confirm in their Methods section that their experiments were carried out according to the guidelines laid down by their institution's animal welfare committee, including an ethics approval reference number. The Methods section must contain a statement about access to food, water and housing, details of the anaesthetic regime: anaesthetic used, dose and route of administration, and method of killing the experimental animals.

Specifically, please include details of the food, water and housing for the animals, along with details of the anaesthesia (agent, route, dose) used during ex-vivo experiments, before cervical dislocation.

3) - Your manuscript must include a complete Additional Information section, including the heading 'Additional Information' and subheadings competing interests; funding; author contributions and acknowledgements.

4) - Papers must comply with the Statistics Policy: https://jp.msubmit.net/cgi-bin/main.plex?form_type=display_requirements#statistics.

In summary:

- If $n \leq 30$, all data points must be plotted in the figure in a way that reveals their range and distribution. A bar graph with data points overlaid, a box and whisker plot or a violin plot (preferably with data points included) are acceptable formats.
- If $n > 30$, then the entire raw dataset must be made available either as supporting information, or hosted on a not-for-profit repository, e.g. FigShare, with access details provided in the manuscript.
- 'n' clearly defined (e.g. x cells from y slices in z animals) in the Methods. Authors should be mindful of pseudoreplication.
- All relevant 'n' values must be clearly stated in the main text, figures and tables.
- The most appropriate summary statistic (e.g. mean or median and standard deviation) must be used. Standard Error of the

Mean (SEM) alone is not permitted.

- Exact p values must be stated. Authors must not use 'greater than' or 'less than'. Exact p values must be stated to three significant figures even when 'no statistical significance' is claimed.

5) - Please include an Abstract Figure file, as well as the Figure Legend text within the main article file. The Abstract Figure is a piece of artwork designed to give readers an immediate understanding of the research and should summarise the main conclusions. If possible, the image should be easily 'readable' from left to right or top to bottom. It should show the physiological relevance of the manuscript so readers can assess the importance and content of its findings. Abstract Figures should not merely recapitulate other figures in the manuscript. Please try to keep the diagram as simple as possible and without superfluous information that may distract from the main conclusion(s). Abstract Figures must be provided by authors no later than the revised manuscript stage and should be uploaded as a separate file during online submission labelled as File Type 'Abstract Figure'. Please also ensure that you include the figure legend in the main article file. All Abstract Figures should be created using BioRender. Authors should use The Journal's premium BioRender account to export high-resolution images. Details on how to use and access the premium account are included as part of this email.

6) - Please ensure that all figures and tables have a title and legend, and that they have been cited within the main article text.

EDITOR COMMENTS

Reviewing Editor:

Thank you for submitting your manuscript to The Journal of Physiology. Your work has now been reviewed, and while both Referees acknowledged the technical quality and value of the data, several important concerns were raised.

The most critical issue relates to the substantial similarity between this manuscript and your recent publication in PNAS (Underhill et al., 2025). Although your current work uses a different genetic model, the overlap with the previous study extends beyond general background or hypothesis. There is some duplication in the text - particularly in the Methods section - and the figures follow a very similar structure and layout, including the number and style of subpanels. Additionally, certain parts of the manuscript present data and conclusions that closely mirror those already reported in the earlier study.

In light of these issues, substantial revisions are required. The novel contributions of the present work should be clearly articulated, with a specific explanation of how they build upon or differ from the previous findings. Text reused from earlier publications should be rephrased or omitted, particularly in the Methods section, and any reuse of figure layouts must be limited to cases where it is clearly warranted and appropriately justified.

The manuscript should also adhere to The Journal of Physiology's data reporting and statistics policy - with regard to consistent reporting of sample sizes, statistical tests, measures of variability, and exact p-values (particular attention should be given to Figure 7). Please also include a description of the animal housing conditions and access to food and water.

In addition to the above issues, the Referees have outlined several additional concerns that should be thoroughly addressed.

Senior Editor:

Thank you for submitting your work to JP. We are willing to entertain a revision, but were extremely concerned by the extent to which you have recycled text verbatim from your prior publications without disclosing that this was done. While it can be difficult to find different ways to say something, particularly in the Methods section, publication ethics require that you must make every effort to do so. Thus, we expect you to thoroughly revise your manuscript if you wish it to be considered further.

REFEREE COMMENTS

Referee #1:

Myo7a is one of the causative genes for syndromic or non-syndromic deafness. The gene product is believed to be a part of tip-link complex and to regulate resting open probability (P₀) of MET channel in the hair cell. Although the analysis of Myo7a conditional knockouts has revealed that Myo7a is required for maintaining functional and structural integrity of hair bundles (Underhill et al., 2025 PNAS), the significance of Myo7a in the immature hair cells remained to be fully understood. Using Shaker1 mice carrying missense mutation on Myo7a gene, this follow-up study conducted by the same group demonstrated that Myo7a regulates resting P₀ in immature hair cells but not in mature cells. Treatment of mutant cochleae with M β CD, a reagent that alters the lipid composition of plasma membrane, restored resting P₀ to the normal level, suggesting indirect

regulation of P0 by Myo7a in immature hair cells.

The novelty of findings in this study is limited, but the results are explicitly stated and discussed thoroughly. In addition, although in general the measurement of MET current by patch clamp is technically difficult, this experiment was elegantly conducted by well-known experts in this work. Thus, the data are reliable. Overall, the findings provide new insights into mechanical properties of hair bundle.

1. I was just wondering whether hair cells from Myo7a^{+/+} mice have the similar hair bundle properties to those of Myo7a^{ash1/+}.

2. Fig.7F, Up-regulated genes are shown as Down-regulated.

Referee #2:

This paper is a companion to one published this year in PNAS. It has an overlapping set of authors and the overall style follows that paper very closely albeit with a different mouse mutation. The sequence of figures and even the Methods are essentially identical (for which see below). The 2025 PNAS paper concludes, based on a conditional gene knock out at mouse stage P5, that a function of the protein MYO7A is to maintain the cochlear hair cell bundle into adulthood. This is an interesting result since given the three decades since the identification of the Shaker-1 mutated protein - really the beginning of auditory molecular biology - the precise roles of MYO7A have been unclear. It is further complicated by the cytoplasmic expression of the protein clearly suggesting multiple functions in hair cells.

The present paper uses the classic Shaker-1 mutation and uses it to see how the effects of that MYO7A mutation change from P5-P10. Despite the beguiling title the running head reveals this as intended as a developmental paper. There are multiple messages however and the title's claim that MYO7A 'regulates' the transducing stereocilia (how? No mechanism is really explored) is very different from the message carried by the running head.

As a paper suitable for J Physiol it is overwritten: there are 80+ references. It would be much stronger with some serious pruning and focus. In particular there is a lot of introductory background material in the Results section which could well be slimmed or put elsewhere. Not least, the use of multiple (up to 14) subpanels in figures makes the information and arguments very hard to follow. These are bad habits.

It is shown by s.e.m. that the shaker-1 homozygote mutation leads to shortened row2 and 3 stereocilia by P10. That the two shorter transducing rows of stereocilia in both inner and outer hair cells become shorter is novel, even though the normal staircase structure of the bundle is well established by P5. This is interesting and to the best of my knowledge has not been pointed out before but there is little quantification of this effect - it could be valuable; the connection with the stiffness and bundle repositioning in described in the 2025 PNAS paper Fig 4 seems to have been left undeveloped.

It is also shown that between P5 and P 10 that the shaker-1 mutation leads to reduced transduction currents - as might be anticipated given the structural observations - but decoupled from changes in the resting open state of the channel and its calcium dependence (using experiments with BAPTA in the pipette). The 'rescue' of the closed transduction channel in the Shaker mutant at P5 by cholesterol depletion is also a surprise, but not taken any further; (the role of lipids in hair cell transduction is really a sideline issue). Nevertheless, Figures 2 and 3 are poorly laid out and tricky to follow for these arguments; the juxtaposition of multiple panels in the layout needs to be improved. It might be worth removing some of the +124/-94 mV data as they clutter the reasoning and are not explicitly referred to in the text.

Methods:

II.152 et seq: There are substantial sections here that are straight cut-and-paste from the supplementary appendices of the 2025 PNAS paper. Copyright (if not worse) issues are involved even though the status of supplementary data in other journals is ambiguous. It might be better to just refer the reader to that paper without resorting to self-plagiarism? Even in these sections there is information of questionable value: e.g. at line 181 what is the value of stating that the length of the fluid pipette is 5.5 cm? or even at lines 196-7 the precise box dimensions other than it is irregular to avoid standing waves.

II. 250 - 257 There is inconsistency between not giving company information for the molecular biology products (Quiagen? BioAnalyzer? for example) and the very detailed information for the electrophysiology and structural imaging given in the previous sections.

Results:

II 306-310 This is the point where the notion of a P_o is introduced. It is buried. It would be more helpful to have an illustrative figure showing how P_o can be determined from the saturating currents produced by a fluid jet.

II.352-365 The paper includes observations that MbCD reverses the effects of the homozygote Shaker-1 mutation. Some further detail is required: 3.5 mM effects are shown; how long was it applied before the data was collected? I. 359 states that other concentrations were used - to what effect? MbCD is a messy drug and there may well have been structural changes as well as the lipids are reorganised. Although interesting there is a shortage of information supporting the result.

II.409-410. Two genes, Mafa and Dnajc5b, are mentioned but there is no follow-up. This should be a Discussion point (see also I.415) rather than left in the Results.

Figures:

The ordering and the style of the figures is really too close for comfort to that of the 2025 PNAS paper. Figures 2 and 3 are extremely crowded to no good effect. Removal of the raw transduction current data, as suggested above, once an example is shown, would be a considerable help. Plotting the Po data as a time-line would also be helpful to show how the changes occur over the period. It would reduce the clutter and cohere useful conclusions from the isolated time point data.

Figure 6 is not helpful. There is no explanation what the program Ballgown (unreferenced) is showing - it is program that may well have a finite lifetime. No for that matter is any attempt made to explain what the two principle components are mainly determined by. I appreciate that there is a growing tendency to present data such as this as revealing mechanism - but the case has not been made here. The figure should be removed.

Minor typo: I. 118. 'dysfunction'

END OF COMMENTS

JP-RP-2025-289623

We thank the Reviewers for their comments, which we have addressed in the revised manuscript. Line numbers refer to: Underhill et al 2025_Revised_Changes Highlighted.pdf

In response to the Editors' and Reviewers' comments, we conducted several additional experiments and expanded our data analysis, which we believe have significantly improved the manuscript. We have also undertaken a major revision of the text to eliminate any partial overlap with our PNAS publication, especially in the Methods section. We trust that these extensive revisions address all concerns raised.

Required Items

1) Author photo and profile.

Included

2) You must start the Methods section with a paragraph headed Ethical approval

Done.

- Research must comply with The Journal's policies regarding animal experiments (<https://physoc.onlinelibrary.wiley.com/hub/animal-experiments>) and adherence to these policies must be stated in the manuscript.

Yes, and policy added to the revised text (ln. 135-137).

- Authors should confirm in their Methods section that their experiments were carried out according to the guidelines laid down by their institution's animal welfare committee, including an ethics approval reference number.

Yes, was already included in the submitted version (ln. 120-122).

- The Methods section must contain a statement about access to food, water and housing, details of the anaesthetic regime: anaesthetic used, dose and route of administration, and method of killing the experimental animals.

The missing information about food and water availability has now been included in the revised manuscript (ln. 1123).

3) Your manuscript must include a complete Additional Information section, including the heading 'Additional Information' and subheadings competing interests; funding; author contributions and acknowledgements.

Yes (Pg. 24).

4) Papers must comply with the Statistics Policy

Yes

5) Please include an Abstract Figure file, as well as the Figure Legend text within the main article file. *Abstract Figures should be uploaded as a separate file during online submission labelled as File Type 'Abstract Figure'. Please also ensure that you include the figure legend in the main article file.*

The Abstract Figure has been added as image file and the legend added at the end of the Figure Legend section.

Reviewing Editor:

Thank you for submitting your manuscript to The Journal of Physiology. Your work has now been reviewed, and while both Referees acknowledged the technical quality and value of the data, several important concerns were raised.

The most critical issue relates to the substantial similarity between this manuscript and your recent publication in PNAS (Underhill et al., 2025). Although your current work uses a different genetic model, the overlap with the previous study extends beyond general background or hypothesis. There is some duplication in the text - particularly in the Methods section - and the figures follow a very similar structure and layout, including the number and style of subpanels. Additionally, certain parts of the manuscript present data and conclusions that closely mirror those already reported in the earlier study.

As requested, we have substantially re-written the Methods and the Results section. While we understand and agreed on the changes in the text, the comment on the Figures is somewhat surprising since as a lab, like other labs, we have a particular 'house style' of doing Figures, which has nothing to do with the quality of the results. We fully agree, however, that some of the Figures contains too many panels, which could make the appreciation of the results more challenging for the reader. To address this issue, we have increased the number of Figures, each showing a specific aspect of the Results. This addresses the comments from Reviewer 2 regarding this point.

This study was performed on a different mouse line (*Shaker-1*), a spontaneous missense mutation resulting in a non-functional protein, compared to that of the previous study (*Myo7a^{fl/fl}Myo15-Cre^{+/-}*), a conditional KO. Because of the different timing of the gene defect, the age range used was also different (developing vs adult cochlea, respectively). Despite this, some of the results and conclusions are indeed qualitative the same, indicating the robustness of our argument that MYO7A is not directly required for tensioning of the tip links. However, in the revised version of this study, we have expanded the results section with new experiments, providing new insights into the potential mechanisms, which were not identified in the previously published PNAS study.

In light of these issues, substantial revisions are required. The novel contributions of the present work should be clearly articulated, with a specific explanation of how they build upon or differ from the previous findings. Text reused from earlier publications should be rephrased or omitted, particularly in the Methods section, and any reuse of figure layouts must be limited to cases where it is clearly warranted and appropriately justified.

As evident from the resubmitted document (with changes highlighted in red), we have substantially revised the manuscript. Both the Methods and Results sections have been largely rewritten. Regarding the figure layout, we believe that our approach provides an effective balance between raw and averaged data. Nonetheless, we have increased the number of figures and reduced the number of panels in some of the figures to further improve clarity.

The manuscript should also adhere to The Journal of Physiology's data reporting and statistics policy - with regard to consistent reporting of sample sizes, statistical tests, measures of variability, and exact p-values (particular attention should be given to Figure 7).

The sample sizes are reported in the Methods section. The additional information is now given in the revised Figure legends.

Please also include a description of the animal housing conditions and access to food and water.

Done

In addition to the above issues, the Referees have outlined several additional concerns that should be thoroughly addressed.

See below for our replies.

Senior Editor:

Thank you for submitting your work to JP. We are willing to entertain a revision, but were extremely concerned by the extent to which you have recycled text verbatim from your prior publications without disclosing that this was done. While it can be difficult to find different ways to say something, particularly in the Methods section, publication ethics require that you must make every effort to do so. Thus, we expect you to thoroughly revise your manuscript if you wish it to be considered further.

As mentioned above, the manuscript has been extensively revised based on the suggestions from both the Reviewing Editor and Referees. We have also performed several additional experiments to strengthen our conclusions.

Referee #1:

Myo7a is one of the causative genes for syndromic or non-syndromic deafness. The gene product is believed to be a part of tip-link complex and to regulate resting open probability (P₀) of MET channel in the hair cell. Although the analysis of Myo7a conditional knockouts has revealed that Myo7a is required for maintaining functional and structural integrity of hair bundles (Underhill et al., 2025 PNAS), the significance of Myo7a in the immature hair cells remained to be fully understood. Using Shaker1 mice carrying missense mutation on Myo7a gene, this follow-up study conducted by the same group demonstrated that Myo7a regulates resting P₀ in immature hair cells but not in mature cells. Treatment of mutant cochleae with MβCD, a reagent that alters the lipid composition of plasma membrane, restored resting P₀ to the normal level, suggesting indirect regulation of P₀ by Myo7a in immature hair cells.

The novelty of findings in this study is limited, but the results are explicitly stated and discussed thoroughly. In addition, although in general the measurement of MET current by patch clamp is technically difficult, this experiment was elegantly conducted by well-known experts in this work. Thus, the data are reliable. Overall, the findings provide new insights into mechanical properties of hair bundle.

Thank you. In the revised version of the manuscript, we have performed several new experiments and data analysis to provide a better understanding of the roles of MYO7A in mechano-electrical transduction. These new data are shown in the new **Figures 10 and 13**, and in new panels adding to existing data (**Figures: 1F, 5C,D, 7F-H, 8F-J**).

1. I was just wondering whether hair cells from Myo7a^{+/+} mice have the similar hair bundle properties to those of Myo7a^{ash1/+}.

Considering the recessive nature of the mutation, the hearing capability of wild-type and heterozygous mice is indistinguishable, which was tested in earlier studies (see our Introduction). Although we do not breed wild-type mice, the structure and dimensions of the hair bundle in heterozygous mice are indistinguishable to those previously reported in isogenic C57BL wild-type mice.

2. Fig.7F, Up-regulated genes are shown as Down-regulated.

Thank you for spotting this error, which we have now rectified in the revised **Figure 12**.

Referee #2:

This paper is a companion to one published this year in PNAS. It has an overlapping set of authors and the overall style follows that paper very closely albeit with a different mouse mutation. The sequence of figures and even the Methods are essentially identical (for which see below). The 2025 PNAS paper concludes, based on a conditional gene knock out at mouse stage P5, that a function of the protein MYO7A is to maintain the cochlear hair cell bundle into adulthood. This is an interesting result since given the three decades since the identification of the Shaker-1 mutated protein - really the beginning of auditory molecular biology - the precise roles of MYO7A have been unclear. It is further complicated by the cytoplasmic expression of the protein clearly suggesting multiple functions in hair cells.

The present paper uses the classic Shaker-1 mutation and uses it to see how the effects of that MYO7A mutation change from P5-P10. Despite the beguiling title the running head reveals this as intended as a developmental paper. There are multiple messages however and the title's claim that MYO7A 'regulates' the transducing stereocilia (how? No mechanism is really explored) is very different from the message carried by the running head.

We agree that the study primarily focuses on developmental stages of hair cells. Because previous studies proposing a role for MYO7A in tip-link tensioning were conducted only in pre-hearing mice, it was essential for us to determine whether this motor protein plays distinct roles in immature (as examined here using *shaker-1* mice) versus mature hair cells (as shown in our PNAS study). Prompted by the Reviewer's comments, we have carried out several additional experiments to further explore the mechanisms linking MYO7A to transduction, which have strengthened our conclusions. This has resulted in the new **Figures 10 and 13**, and new panels adding further insights to existing data (**Figures: 1F, 5C,D, 7F-H, 8F-J**); for details see specific points below.

We have also changed the Title to more accurately reflect the scope and outcomes of the study.

As a paper suitable for J Physiol it is overwritten: there are 80+ references. It would be much stronger with some serious pruning and focus. In particular there is a lot of introductory background material in the Results section which could well be slimmed or put elsewhere. Not least, the use of multiple (up to 14) subpanels in figures makes the information and arguments very hard to follow. These are bad habits.

We have reduced the number of panels in each figure to the minimum necessary to maintain a clear and logical progression of the narrative. Because JPhysiol does not allow supplementary figures, our only option was to increase the total number of figures in the main manuscript.

The large number of references reflects the broad scope of topics and experimental approaches covered in this study, as well as our commitment to acknowledging relevant prior work rather than being overly selective. Nevertheless, we were able to remove 14 references from the original submission.

Regarding the inclusion of brief background information within the Results section, we believe this is important for providing sufficient context and rationale for some of the more technically challenging experiments. We hope that this approach ensures that the article remains accessible to the broad readership of JPhysiol.

It is shown by s.e.m. that the shaker-1 homozygote mutation leads to shortened row2 and 3 stereocilia by P10. That the two shorter transducing rows of stereocilia in both inner and outer hair cells become shorter is novel, even though the normal staircase structure of the bundle is well established by P5. This is interesting and to the best of my knowledge has not been pointed out before but there is little quantification of this effect - it could be valuable; the connection with the

stiffness and bundle repositioning in described in the 2025 PNAS paper Fig 4 seems to have been left undeveloped.

We have now quantified the number and height of the stereocilia at different developmental stages in both OHCs and IHCs, resulting in the addition of several new panels (**Figures 1F, 5C,D, 8F,G**).

We also carried out additional experiments to measure hair bundle displacement and stiffness in IHCs. Since OHCs and IHCs exhibited similar developmental changes, although shifted in time, we focused on IHCs for these measurements. This choice was driven by the fact that bundle displacement can be assessed with greater reliability in developing IHCs, as they are taller and less stiff than those of OHCs. The new data are presented in **Figure 10**.

It is also shown that between P5 and P 10 that the shaker-1 mutation leads to reduced transduction currents - as might be anticipated given the structural observations - but decoupled from changes in the resting open state of the channel and its calcium dependence (using experiments with BAPTA in the pipette). The 'rescue' of the closed transduction channel in the Shaker mutant at P5 by cholesterol depletion is also a surprise, but not taken any further; (the role of lipids in hair cell transduction is really a sideline issue). Nevertheless, Figures 2 and 3 are poorly laid out and tricky to follow for these arguments; the juxtaposition of multiple panels in the layout needs to be improved. It might be worth removing some of the +124/-94 mV data as they clutter the reasoning and are not explicitly referred to in the text.

As mentioned above, we have reduced the number of panels in each Figure to improve clarity and the flow of the narrative. We have also extended the experiments targeting the lipid bilayer by: **1)** testing whether immature IHCs, which show no resting P_o , are similarly affected by cholesterol depletion as immature OHCs (**Figure 8H-J**) and **2)** assessing whether using PAO to prevent the synthesis of another membrane lipid, PIP_2 , produced effects comparable to cholesterol depletion in immature OHCs (**Figure 7F-H**).

Methods:

ll.152 et seq: There are substantial sections here that are straight cut-and-paste from the supplementary appendices of the 2025 PNAS paper. Copyright (if not worse) issues are involved even though the status of supplementary data in other journals is ambiguous. It might be better to just refer the reader to that paper without resorting to self-plagiarism? Even in these sections there is information of questionable value: e.g. at line 181 what is the value of stating that the length of the fluid pipette is 5.5 cm? or even at lines 196-7 the precise box dimensions other than it is irregular to avoid standing waves.

We have substantially revised the Methods section, streamlining it by removing some procedural details and referring to our previously published work where appropriate. However, we believe that clearly describing the specific experimental conditions used in this study remains essential to ensure full reproducibility of the experiments.

ll. 250 - 257 There is inconsistency between not giving company information for the molecular biology products (Quiagen? BioAnalyzer? for example) and the very detailed information for the electrophysiology and structural imaging given in the previous sections.

This has been rectified.

Results:

ll 306-310 This is the point where the notion of a P_o is introduced. It is buried. It would be more helpful to have an illustrative figure showing how P_o can be determined from the saturating currents produced by a fluid jet.

We now include a visual explanation of the method (**Figure 2E**).

11.352-365 The paper includes observations that MbCD reverses the effects of the homozygote Shaker-1 mutation. Some further detail is required: 3.5 mM effects are shown; how long was it applied before the data was collected? l. 359 states that other concentrations were used - to what effect? MbCD is a messy drug and there may well have been structural changes as well as the lipids are reorganised. Although interesting there is a shortage of information supporting the result.

We have expanded the Methods section to provide a clearer explanation of the experimental approach used with M β CD and with the newly introduced compound PAO (**Pg. 7**). All points raised by the reviewer have been fully addressed in the revised text. Importantly, as detailed in the updated Methods, all measurements were carried out using drug concentrations and exposure times that allowed us to assess their effects on MET channel open probability before any substantial reduction in MET current occurred. This strategy should minimise, or avoid, potential off-target or secondary effects of the compounds.

11.409-410. Two genes, Mafa and Dnajc5b, are mentioned but there is no follow-up. This should be a Discussion point (see also 1.415) rather than left in the Results.

We have not followed up these two genes because we could not find a way to test for protein expression with any reliable antibodies. However, we found that the level of ESPNL at the stereocilia tip was significantly reduced in the absence of MYO7A, which has resulted in the new **Figure 13**. This new finding is also followed up in the Discussion.

Figures:

The ordering and the style of the figures is really too close for comfort to that of the 2025 PNAS paper. Figures 2 and 3 are extremely crowded to no good effect. Removal of the raw transduction current data, as suggested above, once an example is shown, would be a considerable help. Plotting the Po data as a time-line would also be helpful to show how the changes occur over the period. It would reduce the clutter and cohere useful conclusions from the isolated time point data.

While we understand and agreed that changes in the text are required, the comment on the Figures is somewhat surprising considering that every lab has a particular 'house style' of making Figures.

We fully agree, however, that some of the Figures contains a large number of panels, and as explained above this issue has been addressed.

We have also added a new Figure summarizing the changes in resting Po with age and when using the different compounds (**Figure 15**), which is now mentioned in the Discussion.

Figure 6 is not helpful. There is no explanation what the program Ballgown (unreferenced) is showing - it is program that may well have a finite lifetime. No for that matter is any attempt made to explain what the two principle components are mainly determined by. I appreciate that there is a growing tendency to present data such as this as revealing mechanism - but the case has not been made here. The figure should be removed.

This Figure is important because it shows that: **1)** the expression level of *Myo7a* is normal in these mice and **2)** only the canonical long isoforms, and no difference in splicing, was found in *Myo7a*^{Sh1/Sh1} mice, which is different from previously published work in conditional *Myo7a* knockout mice (Li et al., 2020). Therefore, we would like to retain the Figure, considering that Supplementary Figures are not allowed in JPhysiol.

We have also revised the Figure legend (now **Figure 11**) with information about the two principal components and the software used, which hopefully addresses your concern.

Minor typo: l. 118. 'dysfunction'

Corrected

Dear Dr Marcotti,

Re: JP-RP-2025-289623R1 **"Myosin 7a is required for maintaining the transducing stereocilia and for force transmission to the MET channel during cochlear hair cell development"** by Anna Underhill, Samuel Webb, Fiorella Carla Grandi, Adam Carlton, Jing-Yi Jeng, Stuart Leigh Johnson, Corne J Kros, and Walter Marcotti

Thank you for submitting your manuscript to The Journal of Physiology. It has been assessed by a Reviewing Editor and by 2 expert referees and we are pleased to tell you that it is acceptable for publication following satisfactory revision.

REVISION CHECKLIST:

Please upload two versions of your manuscript text: one with all relevant changes highlighted and one clean version with no changes tracked. The manuscript file should include all tables and figure legends, but each figure/graph should be uploaded as separate, high-resolution files. The journal is now integrated with Wiley's Image Checking service. For further details, see: <https://www.wiley.com/en-us/network/publishing/research-publishing/trending-stories/upholding-image-integrity-wileys->

image-screening-service

We look forward to receiving your revised submission.

Yours sincerely,

Kim Barrett
Senior Editor
The Journal of Physiology

EDITOR COMMENTS

Reviewing Editor:

Thank you for your revised manuscript and for the considerable effort you have put into addressing the Referees' comments. The Referees agree that the dataset is technically solid and of interest. That said, some concerns remain. We therefore ask that you carefully address the additional comments from the Referees and consider whether further editorial shaping (through streamlining the text and simplifying or consolidating figures) could help better highlight the main conclusions of the study.

Please also clarify in the Statistical Analysis paragraph how the assumptions of parametric testing (including normality of data distribution) were assessed.

REFEREE COMMENTS

Referee #1:

In the revised manuscript, modifications to the figure layout, including the addition of new data, have been made, which strengthen the authors' argument. Furthermore, with the combined efforts of the authors and the other reviewer, the revised version has become more reader-friendly. I have no further concerns aside from the minor comment below.

In the new Fig. 13, ESPNL protein is shown to be primarily localized to the shorter stereocilia. In Myo7ash1/sh1, ESPNL puncta appear not only reduced but also mislocalized to the tips of the first row of the longest stereocilia. If the authors agree with this interpretation, it should be explicitly stated.

Fig. 15 could be presented as a time-course line graph (+ SEM) rather than discrete plotted data to make it easier for readers to understand the overall results (e.g., Myo7a^{sh1/+}, Myo7a^{sh1/sh1}).

Referee #2:

This is a revised paper and many of the comments have been taken on board - thank you. But it has virtually been rewritten and remains as bulky as it was before, perhaps even more so. Although clearly a marker protein, what Myo7a does for hair cells is a problem that has been around for a long time: the paper aggregates a number of approaches of the classic mutation, Shaker, and describes the consequences during development. Does it really pin down how MYO7A contributes mechanistically in a strict sense? The paper adds some new observations but the title itself reveals a degree of uncertainty about what the conclusion really is. That being said the Discussion is good and pulls together this and other results from a wide range of authors in a very helpful way.

The real issue though is that this paper has ended up as a very long read, with clearly multiple contributors over probably an

extended time. If published it has to be slimmed down and to remove some of the more glaring problems. The appeal to a 'house style' to present the data in a particular way seems to me to be unnecessary.

Some points that can be helpfully addressed:

1) The molecular biology Figure 11 should be removed - as in my view it does not provide any insight (even after reading the Frazee et al paper) as a) the PCA analysis is overinterpreted without justification and b) the notation labelling in C remains obscure. Figure 12 is helpful though.

2) Some of the records show currents and stimuli that cannot be correct: the stimulus polarity in Fig1A conflicts with fig1E and for that matter the other Figures. Once shown (and corrected) the stimulus voltage can be removed after being referred to once and omitted from all the other figures (irrespective of a 'house' style!) as indeed happens in some figures.

3) A general comment about the Figure: they are still cluttered, even if no longer presented as multiple panels. Axis labels are repeated across a line unnecessarily and even much of the axis labelling could be simplified - for example P_o (which is the symbol defined, not P_{open}) is all that is required rather than e.g. $P_{open} @ -124 \text{ mV}$; likewise for currents: IT need to in place just once and defined if really required in the legend. The arrows on current records in Figs 9 & 10, for example, seem not be clearly associated with the labels to which they refer.

4) There may be some editing errors. As examples:

a) I.169. Why does the fluid jet contain Cs solution rather than an a normal extracellular solution?! Isn't this the intracellular pipette solution?

b) I.202. There are two microscopes types mentioned on I.152 but not this one.

c) Are all data taken from the cochlear 12 kHz point? - this seems to be only reference (I.221) to the cochlear section used.

d) The Abstract Figure (Fig 16) does seems to imply that outer and inner hair cells look the same, even at this stage of development. This is surely not intended?

END OF COMMENTS

JP-RP-2025-289623R1

We thank the Reviewers for their final additional remarks, which we have addressed in the revised manuscript. Line numbers refer to: Underhill et al 2026_Revised_Changes Highlighted.pdf

Reviewing Editor:

Thank you for your revised manuscript and for the considerable effort you have put into addressing the Referees' comments. The Referees agree that the dataset is technically solid and of interest. That said, some concerns remain. We therefore ask that you carefully address the additional comments from the Referees and consider whether further editorial shaping (through streamlining the text and simplifying or consolidating figures) could help better highlight the main conclusions of the study.

We have addressed the last remaining points (see below).

Please also clarify in the Statistical Analysis paragraph how the assumptions of parametric testing (including normality of data distribution) were assessed.

Thank you for highlighting this point, which we have included in our revised manuscript. We have re-checked all comparisons for normality and we have made a few minor changes to the Figures below, which have been highlighted in the revised Figure legends.

Figure	Shapiro-Wilk test Pass (Yes/No)	Test used	Post test
Figure 1A	No	Mann-Whitney U test	N/A
Figure 1B	No	Aligned ranks transformation two-way ANOVA	N/A
Figure 1C	No		
Figure 7D,E	No	Mann-Whitney U test	N/A
Figure 7G,H	No	Mann-Whitney U test	N/A
Figure 9C,D	No	Mann-Whitney U test	N/A
Figure 12C	No	Mann-Whitney U test	N/A

Importantly, these changes have not affected any of the conclusions. We have also substantially re-written the Statistical section of the Methods (ln. 300-307).

Referee #1:

In the revised manuscript, modifications to the figure layout, including the addition of new data, have been made, which strengthen the authors' argument. Furthermore, with the combined efforts of the authors and the other reviewer, the revised version has become more reader-friendly. I have no further concerns aside from the minor comment below.

Thank you.

In the new Fig. 13, ESPNL protein is shown to be primarily localized to the shorter stereocilia. In *Myo7ash1/sh1*, ESPNL puncta appear not only reduced but also mislocalized to the tips of the first row of the longest stereocilia. If the authors agree with this interpretation, it should be explicitly stated.

Yes, we agree with the reviewer's interpretation. We have now amended the Figure (now Figure 12) and the text (ln. 1159-1161).

Fig. 15 could be presented as a time-course line graph (+ SEM) rather than discrete plotted data to make it easier for readers to understand the overall results (e.g., *Myo7ash1/+*, *Myo7ash1/sh1*).

We agree that the graph was difficult to understand in the format presented. We believe that the new version of the Figure is addressing the point raised (Now Figure 14).

Referee #2:

This is a revised paper and many of the comments have been taken on board - thank you. But it has virtually been rewritten and remains as bulky as it was before, perhaps even more so. Although clearly a marker protein, what Myo7a does for hair cells is a problem that has been around for a long time: the paper aggregates a number of approaches of the classic mutation, Shaker, and describes the consequences during development. Does it really pin down how MYO7A contributes mechanistically in a strict sense? The paper adds some new observations but the title itself reveals a degree of uncertainty about what the conclusion really is. That being said the Discussion is good and pulls together this and other results from a wide range of authors in a very helpful way.

Thank you.

The real issue though is that this paper has ended up as a very long read, with clearly multiple contributors over probably an extended time. If published it has to be slimmed down and to remove some of the more glaring problems. The appeal to a 'house style' to present the data in a particular way seems to me to be unnecessary.

We respectfully acknowledge the Reviewer's concern regarding the length of the manuscript. We would like to clarify that the increased number of figures was primarily a result of complying with the Reviewer's specific request to reduce the panel density in our original figures. While we believe the data presented remain crucial for the interpretation of our findings, we have agreed to remove the previously submitted Figure 11 (see below).

Some points that can be helpfully addressed:

1) The molecular biology Figure 11 should be removed - as in my view it does not provide any insight (even after reading the Frazee et al paper) as a) the PCA analysis is overinterpreted without justification and b) the notation labelling in C remains obscure. Figure 12 is helpful though.

As mentioned above, we have removed Figure 11.

2) Some of the records show currents and stimuli that cannot be correct: the stimulus polarity in Fig 1A conflicts with Fig 1E and for that matter the other Figures. Once shown (and corrected) the stimulus voltage can be removed after being referred to once and omitted from all the other figures (irrespective of a 'house' style!) as indeed happens in some figures.

We assume the Reviewer is referring to the stimulus voltage (DV) in Fig. 2A versus Fig. 2E, as Figure 1 does not display any stimulus voltage traces. The Reviewer is correct; the DV polarity in Panel E is inverted because the stimulus waveform begins with an inhibitory phase in that specific protocol. However, the DV and current traces are displayed correctly in all other figures.

We have decided to retain the DV records in the figures, as they provide essential information regarding timing, amplitude, and polarity. For example, while most recordings begin with an excitatory (positive) stimulus phase, the protocols in Fig. 7 initiate with an inhibitory (negative) phase. Removing the DV traces would obscure these experimental differences and could lead to confusion for the reader.

3) A general comment about the Figure: they are still cluttered, even if no longer presented as multiple panels. Axis labels are repeated across a line unnecessarily and even much of the axis labelling could be simplified - for example P_o (which is the symbol defined, not Popen) is all that is required rather than e.g. Popen @ -124 mV ; likewise for currents: IT need to be in place just once and defined if really required in the legend. The arrows on current records in Figs 9 & 10, for example, seem not to be clearly associated with the labels to which they refer.

We respectfully disagree with the view that the figures are cluttered. The majority of the figures (9 out of 13) contain only 4–8 panels, which is a standard number for this type of data. Regarding the

axis labels, we believe that clear, specific labelling is essential to ensure the figures are self-explanatory.

We have followed the Reviewer's suggestion to replace 'Popen' with 'Po'. However, we have retained the specific voltage indicators (e.g., '@ -124 mV') because Po and IT were measured at different potentials (e.g., -124 mV vs. +96 mV) across different experiments. Omitting these values would introduce ambiguity.

Finally, regarding the comment on arrows in Figures 9 and 10, we wish to clarify that these figures do not contain any arrows in the current (and previous) version of the manuscript; we wonder if the Reviewer might be referring to a different figure. If so, those arrows are clearly referred into the legend and useful to point at specific features not obvious to the general reader.

4) There may be some editing errors. As examples:

a) l.169. Why does the fluid jet contain Cs solution rather than an a normal extracellular solution?! Isn't this the intracellular pipette solution?

Correct. Thank you for spotting this mistake, which we have corrected in the revised ms (ln. 171-172).

b) l.202. There are two microscopes types mentioned on l.152 but not this one.

Added (ln. 149).

c) Are all data taken from the cochlear 12 kHz point? - this seems to be only reference (l.221) to the cochlear section used.

We used the 12 kHz region for immune labelling. For MET current recordings and bundle displacement measurements, we use the 9-12 kHz region. We have now added this information in the revised ms (ln. 164-165).

d) The Abstract Figure (Fig 16) does seems to imply that outer and inner hair cells look the same, even at this stage of development. This is surely not intended?

Thank you for spotting this mistake. We have amended the Figure.

Dear Professor Marcotti,

Re: JP-RP-2026-289623R2 "**Myosin 7a is required for maintaining the transducing stereocilia and for force transmission to the MET channel during cochlear hair cell development**" by Anna Underhill, Samuel Webb, Fiorella Carla Grandi, Adam James Carlton, Jing-Yi Jeng, Stuart Leigh Johnson, Corne J Kros, and Walter Marcotti

We are pleased to tell you that your paper has been accepted for publication in The Journal of Physiology.

Yours sincerely,

Kim Barrett
Senior Editor
The Journal of Physiology

IMPORTANT POINTS TO NOTE FOLLOWING ACCEPTANCE OF YOUR PAPER:

- **IMPORTANT NOTICE ABOUT OPEN ACCESS:** To assist authors whose funding agencies mandate immediate public access to published research findings, The Journal of Physiology allows authors to pay an Open Access (OA) fee to have their papers made freely available immediately on publication.

- You can help your research get the attention it deserves! Check out Wiley's free Promotion Guide for best-practice recommendations for promoting your work at: www.wileyauthors.com/eoo/guide. You can learn more about Wiley Editing Services which offers professional video, design, and writing services to create shareable video abstracts, infographics, conference posters, lay summaries, and research news stories for your research at: www.wileyauthors.com/eoo/promotion.

- If you would like to receive our 'Research Roundup', a monthly newsletter highlighting the cutting-edge research published in The Physiological Society's family of journals (The Journal of Physiology, Experimental Physiology, Physiological Reports, The Journal of Nutritional Physiology and The Journal of Precision Medicine: Health and Disease), please click this link, fill in your name and email address and select 'Research Roundup': <https://www.physoc.org/journals-and-media/membernews>

EDITOR COMMENTS

Reviewing Editor:

Thank you for carefully addressing all of the comments and suggestions. The manuscript has improved substantially as a result, and the revisions have clearly strengthened the work. I appreciate your efforts.

REFEREE COMMENTS

Referee #1:

In the revised manuscript, the authors have adequately addressed my suggestions by modifying the newly added figures. In Figure 13, the mislocalization of the ESPNL protein in inner and outer hair cells of Myo7a mutants is clearly described. In the new Figure 14, the developmental changes in the resting open probability of MET channels in hair cells from control and Myo7a mutant mice are presented more clearly.

Referee #2:

There were only some minor changes but thank you for doing those. The comment in the rebuttal letters are well taken and I think it is the right decision to remove the unhelpful Figure 11. I'm also glad to see the annoying niggle of the stimulus reversal in Figs 2A,B has been corrected.